# GatedMTL: Learning to Share, Specialize, and Prune Representations for Multi-task Learning

## Abstract

Jointly learning multiple tasks with a unified network can improve accuracy and data efficiency while simultaneously reducing computational and memory costs. However, in practice, Multi-task Learning (MTL) is challenging, as optimizing one task objective may inadvertently compromise the performance of another: This is known as *task interference*. A promising direction to mitigate such conflicts between tasks is to allocate task-specific parameters, free from interference, on top of shared features, allowing for positive information transfer across tasks, albeit at the cost of higher computational demands. In this work, we propose a novel MTL framework, **GatedMTL**, to address the fundamental challenges of task interference and computational constraints in MTL. GatedMTL learns the optimal balance between shared and specialized representations under a desired computational constraint. We leverage a learnable gating mechanism allowing each individual task to select and combine channels from its own task-specific features and a shared memory bank of features. Moreover, we regularize the gates to learn the optimal balance between allocating additional task-specific parameters and the model's computational costs. Through extensive empirical evaluations, we demonstrate SoTA results on three MTL benchmarks using convolutional as well as transformer-based backbones on CelebA, NYUD-v2, and PASCAL-Context.

## 1 Introduction

Multi-task learning (MTL) involves a joint optimization process wherein multiple tasks are learned concurrently with a unified architecture. By leveraging the shared information among related tasks, MTL has the potential to improve accuracy and data efficiency. In addition, learning a joint representation reduces the computational and memory costs of the model at inference as visual features relevant to all tasks are computed only once: This is crucial for many real-life applications where a single device is expected to solve multiple tasks simultaneously (e.g. mobile phones, self-driving cars, etc.). Despite these potential benefits, in practice, MTL training is often met with a key challenge known as *negative transfer* or *task interference* Zhao et al. (2018), which refers to the phenomenon where learning of one task negatively impacts the learning of another task in the same unified architecture. While characterizing and solving task interference is an open issue Wang et al. (2019); Royer et al. (2023), there exist two major lines of work to mitigate this problem: **(i)** Multi-task Optimization (MTO) techniques aim to balance the training process of each task, while **(ii)** architectural designs carefully allocate shared and task-specific parameters to reduce interference.

MTO approaches aim to balance the losses/gradients of each task to mitigate the extent of the conflicts in the optimization process of shared features. However, the results may still be compromised if the tasks rely on inherently different visual cues, making sharing parameters difficult: For instance, text detection and face recognition require learning very different texture information and object scales. An alternative and orthogonal research direction is to allocate additional task-specific parameters, on top of shared generic features, to bypass task interference. In particular, recent state-of-the-art methods have proposed task-dependent selection and adaptation of shared features Sun et al. (2020); Guo et al. (2020); Bragman et al. (2019) or to leverage a taskonomy of tasks for architecture design Standley et al. (2020); Zamir et al. (2018). In the former approaches, the dynamic allocation of task-specific features is usually performed one task at a time and solving all tasks still

requires multiple forward passes. Consequently, these methods are often as costly as running a single task model for each task and only save on memory costs.

In contrast, we learn to balance shared and specific features jointly for all tasks, which allows us to predict all task outputs in a single forward pass. As for the architecture design methods, they define entirely separate encoders for tasks which are far from one another in a given precomputed taskonomy of tasks: This rigid sharing pattern prevents task interference between these tasks, but also prohibits any potential beneficial sharing. In comparison, we aim to learn a more fine-grained parameter sharing pattern by jointly learning the task features and how to share parts of their representations. Finally, in practice many MTL solutions focus on accuracy more than computational efficiency: For instance, Vandenhende et al. (2020) propose powerful decoders for dense prediction tasks that dominate the computation cost of the shared encoder backbone. In this work, we jointly optimize for accuracy and computational efficiency by introducing a budget-aware regularization constraint on the learned gate.

To mitigate task interference while controlling computational cost, we propose GatedMTL, which learns the optimal balance between sharing and specializing representations for a given computational budget. In particular, we leverage a shared network branch which is used as a shared memory bank of features for task-specific branches to communicate between each other. This communication is enabled through a learnable gating mechanism which lets each individual task to select channels from either the shared branch representations or their own task-specific ones, in each layer. In this way, GatedMTL enables a more flexible asymmetric sharing scheme among the tasks: While a task may highly benefit and contribute to the shared generic representations, it does not have to use the information provided by other tasks to build its own task-specific features. Finally, given that the gating mechanism is only task-conditional (as compared to input-conditional), the learned gating patterns can be used to prune the unselected channels in the shared and task-specific branches: As a result, GatedMTL collapses to a simpler static architecture at inference time. To further control the computational cost of the resulting inference computational graph, we train the gate with a sparsity objective to match a given computational budget, thus regulating the trade-off between computational cost and multi-task performance. In summary, our contributions are as follows:

- We propose a novel method that learns a multi-task parameter sharing pattern for all tasks jointly, alongside the model features.

- We enable a training mechanism to control the balance between accuracy and inference compute cost: During training, the gates dynamically learn to assign features to either a task-specific or shared branch, until reaching an adjustable target computational budget. This results in a unified architecture that can predict all tasks in a single forward pass while providing a mechanism to approach a desired target computational cost.

- Through extensive empirical evaluations, we report SoTA results consistently on three multi-tasking benchmarks with various convolutional as well as transformer-based backbones. We then further investigate the proposed framework through ablation experiments.

## 2 RELATED WORK

**Multi-task optimization** (MTO) methods aim to automatically balance the different tasks when optimizing shared parameters to maximize average performance. Loss-based methods Kendall et al. (2018); Liu et al. (2022) are usually scalable and adaptively scale task losses based on certain statistics (e.g. task output variance); Gradient-based methods Chen et al. (2018b); Liu et al. (2021); Sener & Koltun (2018); Javaloy & Valera (2021); Chen et al. (2020) are more costly in practice as they require storing a gradient per task, but usually yield higher performance.

**MTL Architectures.** Orthogonal to these optimization methods, several works investigate how to design architectures with optimal parameter sharing across tasks to minimize task interference. For instance Standley et al. (2020); Fifty et al. (2021) identify "task affinities" as a guide to isolate parameters of tasks most likely to interfere with one another. Similarly, Guo et al. (2020) apply neural architecture search techniques to design MTL architectures. However, exploring these architecture search spaces is often a costly process. In contrast, works such as Cross-Stitch Misra et al. (2016), MTAN Liu et al. (2019), Adashare Sun et al. (2020) or MuIT Bhattacharjee et al. (2022) propose to learn the task parameter sharing design alongside the model features. However, most of these works

mainly focus on improving the accuracy of the model while neglecting the computational cost: For instance, Misra et al. (2016); Gao et al. (2020) require a full network per task and improve MTL performance through lightweight adapters across task branches. Liu et al. (2019); Bhattacharjee et al. (2022) use task-specific attention module on top of a shared feature encoder, but the cost of the task-specific decoder heads often dominates the final architecture. Finally, Sun et al. (2020); Wallingford et al. (2022) learns a task-specific gating of model parameters. However, due to the dynamic nature of these works, obtaining all task predictions is computationally inefficient as it requires one forward pass through the model per task. Mixture-of-Experts (MoE) Hazimeh et al. (2021); Fan et al. (2022); Chen et al. (2023); Ma et al. (2018) leverage sparse gating to select a subset of the experts for each input example. Similar to prior dynamic gating methods, MoEs are constrained to solving a single task per forward pass. GatedMTL, in contrast, is designed to solve all tasks simultaneously, a requirement in many real-world practical scenarios. Closest to our work is Bragman et al. (2019), which proposes a probabilistic allocation of convolutional filters as task-specific or shared. However, this design only allows for the shared features to send information to the task-specific branches. In contrast, our gating mechanism allows for information to flow in any direction between the shared and task-specific features, thereby enabling cross-task transfer in every layer.

**Task/Domain-specific Adapters.** Our work also shares similarities with a line of continual learning works which learn lightweight adapters to specialize a set of pretrained shared features to a new task: Mallya et al. (2018) adapts a pretrained deep neural network to multiple tasks by learning a set of per-task sparse masks for the network parameters. Similarly, Berriel et al. (2019) select the most relevant feature channels using learnable gates. Unlike these methods our proposed framework predicts all tasks in a single forward pass, allowing for reduced computational costs.

## 3 METHOD

Given $T$ tasks we aim to learn a flexible allocation of shared and task-specific parameters, while optimizing the trade-off between accuracy and efficiency. Specifically, a GatedMTL model is characterized by task-specific parameters $\Phi_t$ and shared parameters $\Psi$. In addition, discrete gates (with parameters $\alpha$) are trained to only select a subset of the most relevant features in both the shared and task-specific branches, thereby reducing the model's computational cost. Under this formalism, the model and gate parameters are trained end-to-end by minimizing the classical MTL objective:

$$\mathcal{L}(\{\Phi_t\}_{t=1}^T, \Psi, \alpha) = \sum_{t=1}^T \omega_t \, \mathcal{L}_t(X, Y_t; \Phi_t, \Psi, \alpha), \tag{1}$$

where $X$ and $Y_t$ are the input data and corresponding labels for task $t$, $\mathcal{L}_t$ represents the loss function associated to task $t$, and $\omega_t$ are hyperparameter coefficients which allow for balancing the importance of each task in the overall objective. In the rest of the section, we describe how we learn and implement the feature-level routing mechanism characterized by $\alpha$. We focus on convolutional architectures in Section 3.1, and discuss the case of transformer-based models in Appendix E.

### 3.1 LEARNING TO SHARE, SPECIALIZE AND PRUNE

Figure 1 presents an overview of the proposed GatedMTL. Formally, let $\psi^\ell \in \mathbb{R}^{C^\ell \times W^\ell \times H^\ell}$ and $\varphi_t^\ell \in \mathbb{R}^{C^\ell \times W^\ell \times H^\ell}$ represent the shared and task-specific features at layer $\ell$ of our multi-task network, respectively. In each layer $\ell$, the gating module $G_t^\ell$ of task $t$ selects relevant channels from either $\psi^\ell$ and $\varphi_t^\ell$. The output of this hard routing operation yields features $\varphi_t'^\ell$:

$$\varphi_t'^\ell = G_t^\ell(\alpha_t^\ell) \odot \varphi_t^\ell + (1 - G_t^\ell(\alpha_t^\ell)) \odot \psi^\ell, \tag{2}$$

where $\odot$ is the Hadamard product and $\alpha_t^\ell \in \mathbb{R}^{C^\ell}$ denotes the learnable gate parameters for task $t$ at layer $\ell$ and the gating module $G_t^\ell$ outputs a vector in $\{0,1\}^{C^\ell}$, encoding the binary selection for each channel. These intermediate features are then fed to the next task-specific layer to form the features $\varphi_t^{\ell+1} = F(\varphi_t'^\ell; \Phi_t^\ell)$.

Similarly, we construct the shared features of the next layer $\ell+1$ by mixing the previous layer's task-specific feature maps. However, how to best combine these $T$ feature maps is a harder problem than

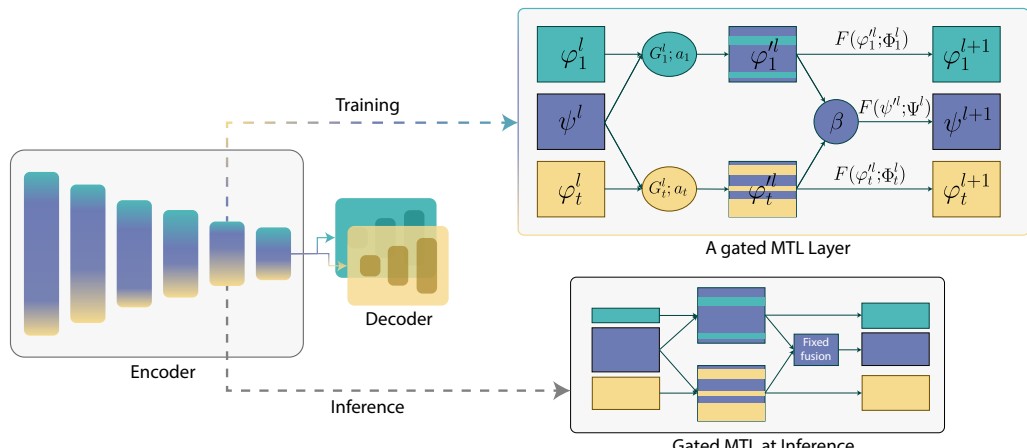

Figure 1: **Overview of the proposed GatedMTL framework**: The original encoder layers are substituted with gated MTL layers. The input to the layer is $t+1$ feature maps, one shared representation and $t$ task-specific representations. To decide between shared $\psi^\ell$ or task-specific $\varphi_t^\ell$ features, each task relies on its own gating module $G_t^\ell$. The resulting channel-mixed feature-map $\varphi_t'^\ell$ is then fed to the next task-specific layer. The input to the shared branch for the next layer is constructed by linearly combining the task-specific features of all tasks using the learned parameter $\beta_t^\ell$. During inference, the features (shared or task-specific) that are not chosen by the gates are removed from the model, resulting in a plain neural network architecture.

the pairwise selection described in (2). Therefore, we let the model learn its own soft combination weights and the resulting mixing operation for the shared features is defined as follows:

$$\psi'^\ell = \sum_{t=1}^T \operatorname*{softmax}_{t=1...T}(\beta_t^\ell) \odot \varphi_t'^\ell, \tag{3}$$

where $\beta^\ell \in \mathbb{R}^{C^\ell \times T}$ denotes the learnable parameters used to linearly combine the task-specific features and form the shared feature map of the next layer. Similar to the task-specific branch, these intermediate features are then fed to a convolutional block to form the features $\psi^{\ell+1} = F(\psi'^\ell; \Psi^\ell)$. Finally, note that there is no direct information flow between the shared features of one layer to the next, i.e., $\psi^\ell$ and $\psi^{\ell+1}$: Intuitively, the shared feature branch can be interpreted as a memory bank through which the task-specific branch can communicate.

**Implementing the discrete routing mechanism.** During **training**, the model features and gates are trained jointly and end-to-end. In (2), the gating modules make a discrete routing decision for each channel: 0 denotes choosing the shared feature, and 1 choosing the specialized feature. In practice, we simply implement $G$ as a sigmoid operation applied to the learnable parameter $\alpha$, followed by a thresholding operation at 0.5. Due to the non-differentiable nature of this operation, we adopt the straight-through estimation (STE) during training Bengio et al. (2013): In the backward pass, STE approximates the gradient flowing through the thresholding operation as the identity function. At **inference**, since the gate modules do not depend on the input data, our proposed GatedMTL method converts to a static neural network architecture, where feature maps are pruned following the learned gating patterns: To be more specific, for a given layer $\ell$ and task $t$, we first collect all channels for which the gate $G_t^\ell(\alpha_t^\ell)$ outputs 0; Then, we simply prune the corresponding task-specific weights in $\Phi_t^\ell$. Similarly, we can prune away weights from the shared branch $\Psi^\ell$ if the corresponding channels are never chosen by any of the tasks in the mixing operation of (2). The pseudo-code for the complete unified encoder forward-pass is detailed in Appendix D.

**Sparsity regularization.** During training, we additionally control the proportion of shared versus task-specific features usage by regularizing the gating module $G$. This allows us to reduce the computational cost, as more of the task-specific weights can be pruned away at inference. We implement the regularizer term as a hinge loss over the gating activations for task-specific features:

$$\mathcal{L}_{\text{sparsity}}(\alpha) = \frac{1}{T} \sum_{t=1}^{T} \max \left( 0, \ \frac{1}{L} \sum_{\ell=1}^{L} \sigma(\alpha_t^\ell) - \tau_t \right), \tag{4}$$

where $\sigma$ is the sigmoid function and $\tau_t$ is a task-specific hinge target. The parameter $\tau$ allows to control the proportion of active gates at each specific layer by setting a soft upper limit for active task-specific parameters. A lower hinge target value encourages more sharing of features while a higher value gives the model the flexibility to select task-specific features albeit at the cost of higher computational costs.

Our final training objective is a combination of the multi-task objective and sparsity regularizer:

$$\mathcal{L} = \mathcal{L}(\{\Phi_t\}_{t=1}^T, \Psi, \alpha, \beta) + \lambda_s \mathcal{L}_{\text{sparsity}}(\alpha), \tag{5}$$

where $\lambda_s$ is a hyperparameter balancing the two losses.

## 4 EXPERIMENTS

### 4.1 EXPERIMENTAL SETUP

**Datasets and Backbones.** We evaluate the performance of GatedMTL on three popular datasets: CelebA Liu et al. (2015), NYUD-v2 Silberman et al. (2012), and PASCAL-Context Chen et al. (2014). CelebA is a large-scale face attributes dataset, consisting of more than 200k celebrity images, each labeled with 40 attribute annotations. We consider the age, gender, and clothes attributes to form three output classification tasks for our MTL setup and use binary cross-entropy to train the model. The NYUD-v2 dataset is designed for semantic segmentation and depth estimation tasks. It comprises 795 train and 654 test images taken from various indoor scenes in New York City. The dataset provides pixel-wise labels for 40 semantic categories. Following recent works Xu et al. (2018); Zhang et al. (2019); Maninis et al. (2019), we also incorporate the surface normal prediction task, obtaining annotations directly from the depth ground truth. We use the mean intersection over union (mIoU) and root mean square error (rmse) to evaluate the semantic segmentation and depth estimation tasks, respectively. We use the mean error (mErr) in the predicted angles to evaluate the surface normals. The PASCAL-Context dataset is an extension of the PASCAL VOC dataset Everingham et al. (2010) and provides a comprehensive scene understanding benchmark by labeling images for semantic segmentation, human parts segmentation, semantic edge detection, surface normal estimation, and saliency detection. The dataset consists of 4,998 train images and 5,105 test images. The semantic segmentation, saliency estimation, and human part segmentation tasks are evaluated using mean intersection over union (mIoU). Similar to NYUD, mErr is used to evaluate the surface normal predictions.

We use ResNet-20 He et al. (2016) as the backbone in our CelebA experiments, with simple linear heads for the task-specific predictions. For the NYUD-v2 dataset, we use ResNet-50 with dilated convolutions and HRNet-18 following Vandenhende et al. (2021). We also present results using a dense prediction transformer (DPT) Ranftl et al. (2021), with a ViT-base and -small backbone. Finally, on PASCAL-Context, we use a ResNet-18 backbone. For both NYUD and PASCAL, we use dense prediction decoders to output the task predictions, as described in Appendix A.

**SotA baselines and Metrics.** We compare GatedMTL to encoder-based methods including Cross-stitch Misra et al. (2016) and MTAN Liu et al. (2019), as well as MTO approaches such as uncertainty weighting Kendall et al. (2018), DWA Liu et al. (2019), and Auto-$\lambda$ Liu et al. (2022), PCGrad Yu et al. (2020), CAGrad Liu et al. (2021), MGDA-UB Sener & Koltun (2018), and RDW Lin et al. (2022). Following Maninis et al. (2019), our main metric is the multi-task performance $\Delta_{\text{MTL}}$ of a model $m$ as the averaged normalized drop in performance w.r.t. the single-task baselines $b$:

$$\Delta_{\text{MTL}} = \frac{1}{T} \sum_{i=1}^{T} (-1)^{l_i} \left( M_{m,i} - M_{b,i} \right) / M_{b,i} \tag{6}$$

where $l_i = 1$ if a lower value means better performance for metric $M_i$ of task $i$, and 0 otherwise. Furthermore, similar to Navon et al. (2022), we compute the mean rank (MR) as the average rank of each method across the different tasks, where a lower MR indicates better performance. All reported results for GatedMTL and baselines are averaged across 3 random seeds.

Finally, to generate the trade-off curve between MTL performance and compute cost of GatedMTL in Figure 2 and all the tables, we sweep over the gate sparsity regularizer weight, $\lambda_s$, in the range of $\{1, 3, 5, 7, 10\} \cdot 10^{-2}$. The task-specific targets $\tau$ in (4) also impact the computation cost: In practice, we use two cues to set the appropriate target values for each task. The first is the gap between the single task performance and the uniform MTL baseline: Intuitively, tasks with significant performance degradation benefit from more task-specific parameters (higher $\tau_t$). Secondly, by studying the distribution of the gating patterns wrt. contribution to the shared branch we can observe which tasks overly share their features at the cost of lower accuracy for themselves. We will analyze the gating patterns for sharing and specialization in section 4.3.2. We further discuss the impact of the sparsity targets $\{\tau_t\}_{t=1}^{T}$ in our ablation experiments in Appendix C.

**Training pipeline.** For initialization, we use pretrained ImageNet weights for the single-task and multi-task baseline. For GatedMTL, the shared branch is initialized with ImageNet weights while the task-specific branches are with their corresponding single-task weights. Finally, we discover that employing a distinct optimizer for the gates improves the convergence of the model. We use SGD with a learning rate of 0.1 for the gates' parameters. We further describe training hyperparameters in Appendix A. All experiments were conducted on a single NVIDIA V100 GPU and we use the same training setup and hyperparameters across all MTL approaches included in our comparison.

## 4.2 Results

In this section, we present the results of GatedMTL, single-task (STL), multi-task baselines, and the competing MTL approaches, on the CelebA, NYUD-v2, and PASCAL-Context datasets. We then present ablation studies on sharing/specialization patterns, the effect of representation capacity of the backbone model on MTL performance, and the application and characterization of the sparsity regularization loss in section 4.3. As mentioned earlier, we also present an ablation on the impact of the sparsity targets $\{\tau_t\}_{t=1}^{T}$ in Appendix C.

### 4.2.1 CelebA

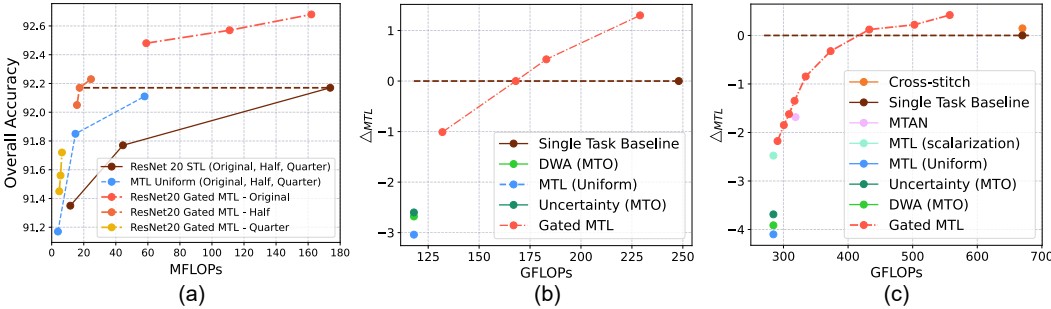

Figure 2: Accuracy vs. floating-point operations (FLOP) trade-off curves for the GatedMTL and SoTA MTL methods. (a) Results on CelebA using ResNet-20 backbone at three different widths (Original, Half, and Quarter). (b) Results on NYUD-v2 using DPT with ViT-small backbone, and (c) Results using ResNet-18 on PASCAL-Context.

Figure 2a shows the trade-off between MTL performance and computational costs (FLOPs) for GatedMTL, MTL uniform, and STL baselines on the CelebA dataset, for 3 different widths for the ResNet-20 backbone: quarter, half, and original capacity. We report the detailed results in Table 8 in the appendix. GatedMTL outperforms MTL uniform and STL baselines with higher overall accuracy at much lower computational costs. Most notably, the performance of GatedMTL with ResNet-20 half width at only 14.8 MFlops matches the performance of STL with 174 MFlops. Finally, we further discuss the behavior of GatedMTL and MTL baselines across different model capacities in the ablation experiment in section 4.3.3.

### 4.2.2 NYUD-v2

Table 1 and 2 present the results on the NYUD-v2 dataset, using the HRNet-18 and ResNet-50 backbones, respectively. As can be seen, most MTL methods improve the accuracy on the segmentation

and depth estimation tasks, while surface normal prediction significantly drops. While MTL uniform and MTO strategies, including DWA, Uncertainty, and Auto-$\lambda$, operate at the lowest computational cost by sharing the full backbone, they fail to compensate for this drop in performance. In contrast, among the encoder-based methods, Cross-stitch largely retains performance on normal estimation and achieves a positive $\Delta_{\text{MTL}}$ score of +1.66. However, this comes at a substantial computational cost, close to that of the STL baseline. In comparison, GatedMTL achieves an overall $\Delta_{\text{MTL}}$ score of +2.06 and +2.04 using HRNet-18 and ResNet-50, respectively, at a lower computational cost.

Table 3 and 4 report the performance of DPT trained models with the ViT-base and ViT-small backbones, and Figure 2b illustrates the trade-off between $\Delta_{\text{MTL}}$ and computational costs of various methods using the ViT-small backbone. The MTL uniform and MTO baselines, display reduced computational costs, yet once again manifesting a performance drop in the normals prediction task. Similar to the trend between HRNet-18 and ResNet-50, the performance drop is more substantial for the smaller model, ViT-small, indicating that task interference is more prominent in small capacity settings. In comparison, GatedMTL consistently demonstrates a more favorable balance between the computational costs and the overall MTL accuracy across varied backbones.

Table 1: Performance comparison on NYUD-v2 using HRNet-18 backbone. Different GatedMTL models are obtained by varying $\lambda_s$.

| Model | Semseg ↑ | Depth ↓ | Normals ↓ | $\Delta_{\text{MTL}}$ (%) ↑ | Flops (G) | Param (M) | MR↓ |
|---|---|---|---|---|---|---|---|
| STL | 41.70 | 0.582 | **18.89** | 0 ± 0.12 | 65.1 | 28.9 | 8.0 |
| MTL (Uni.) | 41.83 | 0.582 | 22.84 | -6.86 ± 0.76 | 24.5 | 9.8 | 11.0 |
| DWA | 41.86 | 0.580 | 22.61 | -6.29 ± 0.95 | 24.5 | 9.8 | 8.7 |
| Uncertainty | 41.49 | 0.575 | 22.27 | -5.73 ± 0.35 | 24.5 | 9.8 | 8.3 |
| Auto-$\lambda$ | 42.71 | 0.577 | 22.87 | -5.92 ± 0.47 | 24.5 | 9.8 | 8.0 |
| RDW | 42.10 | 0.593 | 23.29 | -8.09 ± 1.11 | 24.5 | 9.8 | 11.7 |
| PCGrad | 41.75 | 0.581 | 22.73 | -6.70 ± 0.99 | 24.5 | 9.8 | 10.3 |
| CAGrad | 42.31 | 0.580 | 22.79 | -6.28 ± 0.90 | 24.5 | 9.8 | 8.7 |
| MGDA-UB | 41.23 | 0.625 | 21.07 | -6.68 ± 0.67 | 24.5 | 9.8 | 11.3 |
| GatedMTL | **43.58** | **0.559** | 19.32 | **+2.06** ± 0.13 | 43.2 | 18.8 | **1.3** |
| GatedMTL | 42.95 | 0.562 | 19.73 | +0.68 ± 0.09 | 38.3 | 16.5 | 2.3 |
| GatedMTL | 42.36 | 0.564 | 20.04 | -0.55 ± 0.17 | 36.0 | 15.4 | 4.0 |
| GatedMTL | 42.73 | 0.575 | 21.01 | -2.55 ± 0.11 | 33.1 | 13.7 | 4.0 |
| GatedMTL | 42.35 | 0.575 | 21.70 | -4.07 ± 0.38 | 29.2 | 11.9 | 5.7 |

Table 2: Performance comparison on NYUD-v2 using ResNet-50 backbone. Different GatedMTL models are obtained by varying $\lambda_s$.

| Model | Semseg ↑ | Depth ↓ | Normals ↓ | $\Delta_{\text{MTL}}$ (%) ↑ | Flops (G) | Param (M) | MR↓ |
|---|---|---|---|---|---|---|---|
| STL | 43.20 | 0.599 | **19.42** | 0 ± 0.11 | 1149 | 118.9 | 9.0 |
| MTL (Uni.) | 43.39 | 0.586 | 21.70 | -3.04 ± 0.79 | 683 | 71.9 | 9.7 |
| DWA | 43.60 | 0.593 | 21.64 | -3.16 ± 0.39 | 683 | 71.9 | 9.7 |
| Uncertainty | 43.47 | 0.594 | 21.42 | -2.95 ± 0.40 | 683 | 71.9 | 10.0 |
| Auto-$\lambda$ | 43.57 | 0.588 | 21.75 | -3.10 ± 0.39 | 683 | 71.9 | 10.0 |
| RDW | 43.49 | 0.587 | 21.54 | -2.74 ± 0.09 | 683 | 71.9 | 8.3 |
| PCGrad | 43.74 | 0.588 | 21.55 | -2.66 ± 0.15 | 683 | 71.9 | 7.3 |
| CAGrad | 43.57 | 0.583 | 21.55 | -2.49 ± 0.11 | 683 | 71.9 | 7.0 |
| MGDA-UB | 42.56 | 0.586 | 21.76 | -3.83 ± 0.17 | 683 | 71.9 | 11.3 |
| MTAN | **44.92** | 0.585 | 21.14 | -0.84 ± 0.32 | 683 | 92.4 | 4.0 |
| Cross-stitch | 44.19 | 0.577 | 19.62 | +1.66 ± 0.09 | 1151 | 119.0 | 2.7 |
| GatedMTL | 44.38 | **0.576** | 19.50 | **+2.04** ± 0.07 | 916 | 95.4 | **1.7** |
| GatedMTL | 43.63 | 0.577 | 19.66 | +1.16 ± 0.10 | 892 | 92.4 | 3.7 |
| GatedMTL | 43.05 | 0.589 | 19.95 | -0.50 ± 0.05 | 794 | 83.3 | 9.7 |

Table 3: Performance comparison on NYUD-v2 using DPT with ViT-base. Different GatedMTL models are obtained by varying $\lambda_s$.

| Model | Semseg ↑ | Depth ↓ | Normals ↓ | $\Delta_{\text{MTL}}$ (%) ↑ | Flops (G) | MR↓ |
|---|---|---|---|---|---|---|
| STL | 51.65 | 0.548 | **19.04** | 0 | 759 | 5.0 |
| MTL (Uni.) | 51.38 | 0.539 | 20.73 | -2.57 | 294 | 7.3 |
| DWA | 51.66 | 0.536 | 20.98 | -2.66 | 294 | 6.0 |
| Uncertainty | 51.87 | 0.5352 | 20.72 | -2.02 | 294 | 4.0 |
| GatedMTL | **51.98** | **0.528** | 19.10 | **+1.32** | 626 | **1.3** |
| GatedMTL | 51.46 | 0.536 | 19.34 | +0.08 | 483 | 5.0 |
| GatedMTL | 51.66 | 0.534 | 20.16 | -1.10 | 387 | 3.7 |
| GatedMTL | 51.71 | 0.535 | 20.38 | -1.51 | 324 | 3.7 |

Table 4: Performance comparison on NYUD-v2 using DPT with ViT-small. Different GatedMTL models are obtained by varying $\lambda_s$.

| Model | Semseg ↑ | Depth ↓ | Normals ↓ | $\Delta_{\text{MTL}}$ (%) ↑ | Flops (G) | MR↓ |
|---|---|---|---|---|---|---|
| STL | **46.58** | 0.583 | 21.22 | 0 | 248 | 4.0 |
| MTL (Uni.) | 45.32 | 0.576 | 22.86 | -3.04 | 118 | 7.3 |
| DWA | 45.74 | 0.5721 | 22.94 | -2.68 | 118 | 5.7 |
| Uncertainty | 45.67 | 0.5737 | 22.80 | -2.60 | 118 | 5.7 |
| GatedMTL | 45.96 | **0.5648** | **20.77** | **+1.30** | 229 | **1.7** |
| GatedMTL | 45.34 | 0.5671 | 20.96 | +0.43 | 183 | 4.0 |
| GatedMTL | 45.57 | 0.5666 | 21.36 | +0.00 | 168 | 4.0 |
| GatedMTL | 45.99 | 0.5713 | 22.02 | -1.01 | 132 | 3.7 |

### 4.2.3 PASCAL-CONTEXT

Table 5 summarizes the results of our experiments on the PASCAL-context dataset encompassing five tasks. Note that following previous work, we use the task losses' weights $\omega_t$ from Maninis et al. (2019) for all MTL methods, but also report MTL uniform results as reference. Figure 2c illustrates the trade-off between $\Delta_{\text{MTL}}$ and the computational cost of all models. The STL baseline outperforms most methods on the semantic segmentation and normals prediction tasks with a score of 14.70 and 66.1, while incurring a computational cost of 670 GFlops. Among the baseline MTL and MTO approaches, there is a notable degradation in surface normal prediction. Finally, as witnessed in prior works Maninis et al. (2019); Vandenhende et al. (2020); Brüggemann et al. (2021), we observe that most MTL and MTO baselines struggle to reach STL performance. Among competing methods, MTAN and MTL yield the best MTL performance versus computational cost trade-off, however, both suffer from a notable decline in normals prediction performance.

At its highest compute budget (no sparsity loss and negligible computational savings), GatedMTL outperforms the STL baseline, notably in Saliency and Human parts prediction tasks, and achieves an overall $\Delta_{\text{MTL}}$ of +0.56. As we reduce the computational cost by increasing the sparsity loss

Table 5: Performance comparison on PASCAL-Context. Different GatedMTL models are obtained by varying $\lambda_s$.

| Model | Semseg ↑ | Normals ↓ | Saliency ↑ | Human ↑ | Edge ↓ | $\Delta_{\text{MTL}}$(%) ↑ | Flops (G) | MR↓ |
|---|---|---|---|---|---|---|---|---|
| STL | 66.1 | 14.70 | 0.661 | 0.598 | 0.0175 | 0 | 670 | 5.8 |
| MTL (uniform) | 65.8 | 17.03 | 0.641 | 0.594 | 0.0176 | -4.14 | 284 | 11.6 |
| MTL (Scalar) | 64.3 | 15.93 | 0.656 | 0.586 | 0.0172 | -2.48 | 284 | 10.0 |
| DWA | 65.6 | 16.99 | 0.648 | 0.594 | 0.0180 | -3.91 | 284 | 11.4 |
| Uncertainty | 65.5 | 17.03 | 0.651 | 0.596 | 0.0174 | -3.68 | 284 | 9.8 |
| PCGrad | 62.6 | 15.35 | 0.645 | 0.596 | 0.0174 | -2.58 | 284 | 11.4 |
| CAGrad | 62.3 | 15.30 | 0.648 | 0.604 | 0.0174 | -2.03 | 284 | 9.6 |
| MGDA-UB | 63.0 | 15.34 | 0.646 | 0.604 | 0.0174 | -1.94 | 284 | 9.6 |
| Cross-stitch | **66.3** | 15.13 | **0.663** | 0.602 | 0.0171 | +0.14 | 670 | 3.8 |
| MTAN | 65.1 | 15.76 | 0.659 | 0.590 | **0.0170** | -1.78 | 319 | 8.4 |
| GatedMTL | 65.7 | 14.71 | **0.663** | **0.606** | 0.0172 | **+0.56** | 664 | **3.0** |
| GatedMTL | 65.1 | **14.64** | **0.663** | 0.604 | 0.0172 | +0.42 | 577 | 4.4 |
| GatedMTL | 65.2 | 14.75 | **0.663** | 0.600 | 0.0172 | +0.12 | 435 | 5.0 |
| GatedMTL | 64.9 | 14.72 | 0.658 | 0.596 | 0.0172 | -0.28 | 377 | 7.0 |
| GatedMTL | 65.1 | 15.02 | 0.655 | 0.592 | 0.0172 | -0.85 | 334 | 8.2 |

Table 6: Comparing the MTL performance using the $L_1$ Hinge loss and the standard $L_1$ loss on PASCAL-Context. Different GatedMTL models are obtained by varying $\lambda_s$.

| Model | $\mathcal{L}_{\text{sparsity}}$ | Semseg ↑ | Normals ↓ | Saliency ↑ | Human ↑ | Edge ↓ | $\Delta_{\text{MTL}}$(%) ↑ | Flops (G) | MR↓ |
|---|---|---|---|---|---|---|---|---|---|
| GatedMTL | None | 65.7 | 14.71 | 0.663 | 0.606 | 0.0172 | +0.56 | 664 | 1.8 |
| GatedMTL | $L_1$ | 63.9 | 14.74 | 0.664 | 0.600 | 0.0172 | -0.27 | 623 | 2.8 |
| GatedMTL | $L_1$ | 61.1 | 15.07 | 0.663 | 0.582 | 0.0172 | -2.20 | 518 | 4.0 |
| GatedMTL | Hinge | 65.1 | 14.64 | 0.648 | 0.604 | 0.0171 | +0.28 | 557 | 2.2 |
| GatedMTL | Hinge | 65.2 | 14.75 | 0.644 | 0.600 | 0.0172 | -0.13 | 433 | 3.4 |

weight $\lambda_s$, we observe a graceful decline in the multi-task performance that outperforms competing methods; This emphasizes our model's ability to maintain a favorable balance between compute costs and multi-task performance across computational budgets.

## 4.3 ABLATION STUDIES

### 4.3.1 SPARSITY LOSS

To study the effect of the sparsity loss defined in Equation 4, we conduct the following two experiments: First, we omit the sparsity regularization loss ($\lambda_s = 0$): As can be seen in the first row of Table 6, GatedMTL is on par with a single task, but the computational savings are very limited. In the second ablation experiment, we compare the use of the $L_1$ hinge loss with a standard $L_1$ loss function as the sparsity regularizer: The results of Table 6 show that the hinge loss formulation consistently yields better trade-offs.

### 4.3.2 LEARNED SHARING AND SPECIALIZATION PATTERNS

We now investigate the gating patterns that the model converges to. Specifically, we want to observe how much each task contributes to and benefits from the shared representations. To that aim, we monitor **(i)** the percentage of task-specific representations selected by each task (captured by the gates $G_t(\alpha_t)$), as well as **(ii)** how much the features specific to each task contribute to the formation of the shared feature bank (captured by the learned combination weights $\beta$); We visualize both of these metrics for the five tasks of the Pascal-Context dataset in Figure 3, for a subset of layers and for different sparsity regularizers: with hinge loss (left), with $L_1$ loss at medium (middle) and high pruning levels (right). We also report these results for all layers in Figure 4 in the Appendix.

In all settings, the semantic segmentation task makes the largest contribution to the shared branch, followed by the normals prediction tasks. It is worth noting that the amount of feature contribution to the shared branch can also be largely influenced by other tasks' loss functions. In this situation, we observe that if the normal task lacks enough task-specific features (as seen in the middle and right models), its performance deteriorates significantly. In contrast, when it acquires sufficient task-specific features, it maintains a high accuracy (left). Intriguingly, the features of the normal task become less interesting to other tasks in this scenario possibly due to increased specialization.

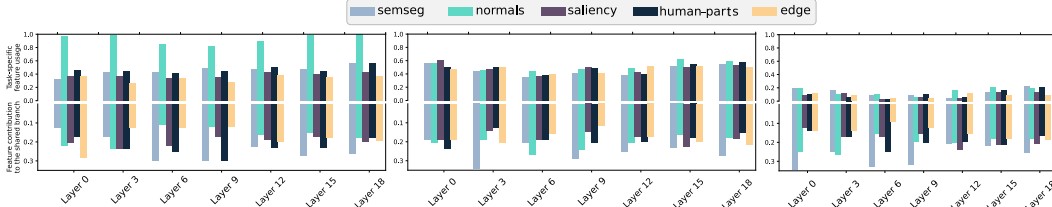

Figure 3: The task-specific representation selection ratio (top) versus proportions of maximum contributions to the shared branch (bottom) for GatedMTL with hinge loss (left), $L_1$ loss with medium pruning (middle) and $L_1$ loss with high pruning (right).

### 4.3.3 IMPACT OF MODEL CAPACITY

In this section, we conduct an ablation study to analyze the relationship between model capacity and multi-task performance. We progressively reduce the width of ResNet-50 and ResNet-20 to half and a quarter of the original sizes for NYUD-v2 and CelebA datasets, respectively. Shrinking the model size, as observed in Table 7, incurs progressively more harmful effect on multi-task performance compared to the single task baseline. In comparison, our proposed GatedMTL approach consistently finds a favorable trade-off between capacity and performance and improves over single task performances, across all capacity ranges.

Table 7: Performance across various model capacities using the ResNet-20 and ResNet-50 backbones on the CelebA (left) and NYUD-v2 (right) tasks.

| | Model | Gender ↑ | Age ↑ | Clothes ↑ | Overall ↑ | Flops (M) | MR↓ |
|---|---|---|---|---|---|---|---|
| Original | STL | 97.50 | 86.02 | **93.00** | 92.17 | 174 | 2.0 |
| | MTL | 97.28 | 86.70 | 92.35 | 92.11 | 58 | 2.7 |
| | GatedMTL | **97.60** | **87.44** | 92.40 | **92.48** | 59 | **1.3** |
| Half | STL | 96.99 | 85.60 | **92.72** | 91.77 | 44.4 | 2.3 |
| | MTL | 97.02 | 86.41 | 92.11 | 91.85 | 14.8 | 2.0 |
| | GatedMTL | **97.33** | **86.75** | 92.05 | **92.05** | 15.5 | **1.7** |
| Quarter | STL | 96.64 | 85.22 | **92.19** | 91.35 | 11.6 | 2.0 |
| | MTL | 96.46 | 85.46 | 91.59 | 91.17 | 3.9 | 2.3 |
| | GatedMTL | **96.81** | **86.05** | 91.48 | **91.45** | 4.7 | **1.7** |

| | Model | Semseg ↑ | Depth ↓ | Normals ↓ | $\Delta_{MTL}$ (%) ↑ | Flops (G) | MR↓ |
|---|---|---|---|---|---|---|---|
| Original | STL | 43.20 | 0.599 | **19.42** | 0 | 1149 | 2.3 |
| | MTL | 43.39 | 0.586 | 21.70 | -3.02 | 683 | 2.3 |
| | GatedMTL | **43.63** | **0.577** | 19.66 | **+1.16** | 892 | **1.3** |
| Half | STL | 39.72 | 0.613 | **20.06** | 0 | 415 | 2.3 |
| | MTL | 40.20 | 0.610 | 22.78 | -3.98 | 296 | 2.0 |
| | GatedMTL | 39.78 | **0.591** | 20.41 | **+0.63** | 348 | **1.7** |
| Quarter | STL | 35.44 | 0.654 | **21.21** | 0 | 177 | 2.3 |
| | MTL | 35.68 | 0.632 | 24.57 | -4.06 | 147 | 2.3 |
| | GatedMTL | **35.71** | **0.624** | 21.75 | **+0.94** | 164 | **1.3** |

## 5 DISCUSSION AND CONCLUSION

In this paper, we proposed GatedMTL, a novel framework to address the fundamental challenges of task interference and computational constraints in MTL. GatedMTL leverages a learnable gating mechanism that facilitates individual tasks to select and combine channels from both specialized and shared feature sets. This framework promotes an asymmetric flow of information, facilitating varied contributions from individual tasks to the shared branch's representations. By regularizing the learnable gates, we can strike a balance between task-specific resource allocation and overarching computational costs. GatedMTL demonstrates state-of-the-art performance across various architectures and on notable benchmarks such as CelebA, NYUD-v2, and Pascal-Context. The gating mechanism in GatedMTL operates over the channel dimension, or the embedding dimension in the case of ViTs, which in its current form does not support structured pruning for attention matrix computations. Future explorations might integrate approaches like patch-token gating to further optimize computational efficiency.

**Limitations.** In this work, we primarily focused on enhancing the trade-off between accuracy and efficiency during inference. However, GatedMTL comes with a moderate increase in training time, along with the prerequisite of having pre-trained models for individual tasks. Additionally, our method is not particularly designed to scale to an extremely large number of tasks. Furthermore, although both $\lambda_s$ and $\tau_t$ can control the trade-off between performance and computational cost, effectively approximating the desired FLOPs, we still cannot guarantee a specific target FLOP. Lastly, we mainly explored the parameter-sharing mechanisms in the model encoder, deferring the exploration of sharing strategies in the decoder to future research.

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

APPENDIX

# A  IMPLEMENTATION DETAILS

## A.1  CELEBA

On the CelebA dataset, we use ResNet-20 as our backbone with three task-specific linear classifier heads, one for each attribute. We resize the input images to 32x32 and remove the initial pooling in the stem of ResNet to accommodate the small image resolution. For training, we use the Adam optimizer with a learning rate of 1e-3, weight decay of 1e-4, and a batch size of 128. For learning rate decay, we use a step learning rate scheduler with step size 20 and a multiplicative factor of $1/3$.

## A.2  NYUD AND PASCAL-CONTEXT

For both NYUD-v2 and PASCAL-Context with ResNet-18 and ResNet-50 backbones, we use the Atrous Spatial Pyramid Pooling (ASPP) module introduced by Chen et al. (2018a) as task-specific decoders. For the HRNet-18 backbone, we follow the methodology of the original paper Wang et al. (2020): HRNet combines the output representations at four different resolutions and fuses them using 1x1 convolutions to output dense prediction.

We train all convolution-based encoders on the NYUD-v2 dataset for 100 epochs with a batch size of 4 and on the PASCAL-Context dataset for 60 epochs with a batch size of 8. We use the Adam optimizer to train all of the models, with a learning rate of 1e-4 and weight decay of 1e-4. We use the same data augmentation strategies for both NYUD-v2 and PASCAL-Context datasets as described in Vandenhende et al. (2020).

In terms of task objectives, we use the cross-entropy loss for semantic segmentation and human parts, $L_1$ loss for depth and normals, and binary-cross entropy loss for edge and saliency detection tasks, similar to Vandenhende et al. (2020). For learning rate decay, we adopt a polynomial learning rate decay scheme with a power of 0.9.

**The Choice of $\omega_t$.** The hyper-parameter $\omega_t$ denotes the scalarization weights. We use the weights suggested in prior work but also report numbers of uniform scalarization. For NYUD-v2, we use uniform scalarization as suggested in Maninis et al. (2019); Vandenhende et al. (2021), and for PASCAL-Context, we similarly use the weights suggested in Maninis et al. (2019) and Vandenhende et al. (2021).

## A.3  DPT TRAINING

For DPT training, we follow the same training procedure as described by the authors, which employs the Adam optimizer, with a learning rate of 1e-5 for the encoder and 1e-4 for the decoder, and a batch size of 8.

The ViT backbones were pre-trained on ImageNet-21k at resolution 224×224, and fine-tuned on ImageNet 2012 at resolution 384×384. The feature dimension for DPT's decoder was reduced from 256 to 64. We conducted a sweep over a set of weight decay values and chose 1e-6 as the optimal value for our DPT experiments.

# B  ADDITIONAL EXPERIMENTS

## B.1  FULL RESULTS ON CELEBA

In Table 8 we report results on the CelebA dataset for different model capacities: Here, GatedMTL is compared to the STL and standard MTL methods with different model width: at original, half and quarter of the original model width.

Table 8: Performance comparison of various MTL models on the CelebA dataset with different model capacities. Different GatedMTL models are obtained by varying $\lambda_s$.

| | Model | Gender ↑ | Age ↑ | Clothes ↑ | Overall ↑ | Flops (M) | MR↓ |
|---|---|---|---|---|---|---|---|
| Original | STL | 97.50 | 86.02 | **93.00** | 92.17 | 174 | 3.3 |
| | MTL | 97.28 | 86.70 | 92.35 | 92.11 | 58 | 4.7 |
| | GatedMTL | 97.60 | 87.44 | 92.40 | 92.48 | 59 | 2.7 |
| | GatedMTL | 97.77 | 87.39 | 92.56 | 92.57 | 11 | 2.3 |
| | GatedMTL | **97.95** | **87.24** | 92.85 | **92.68** | 162 | **2.0** |
| Half | STL | 96.99 | 85.60 | **92.72** | 91.77 | 44.4 | 3.7 |
| | MTL | 97.02 | 86.41 | 92.11 | 91.85 | 14.8 | 4.0 |
| | GatedMTL | 97.33 | 86.75 | 92.05 | 92.05 | 15.5 | 3.7 |
| | GatedMTL | 97.33 | **87.05** | 92.12 | 92.17 | 17.4 | 2.3 |
| | GatedMTL | **97.46** | 86.97 | 92.47 | **92.23** | 24.5 | **1.7** |
| Quarter | STL | 96.64 | 85.22 | **92.19** | 91.35 | 11.6 | 3.3 |
| | MTL | 96.46 | 85.46 | 91.59 | 91.17 | 3.9 | 4.3 |
| | GatedMTL | 96.81 | 86.05 | 91.48 | 91.45 | 4.7 | 3.7 |
| | GatedMTL | **96.92** | 86.10 | 91.64 | 91.56 | 5.5 | **2.0** |
| | GatedMTL | 96.81 | **86.61** | 91.74 | **91.72** | 6.4 | **2.0** |

## B.2 SHARING/SPECIALIZATION PATTERNS

Figure 4 illustrates the distribution of the gating patterns across all layers of the ResNet-18 backbone for the PASCAL-Context dataset for 3 models using (a) a Hinge loss, (b) a medium-level pruning using uniform $L_1$ loss and (c) a high-level pruning with uniform $L_1$ loss.

## C   ABLATION: SPARSITY TARGETS

By tuning the sparsity targets $\tau$ in Equation 4, we can achieve specific compute budgets of the final network at inference. However, there are multiple choices of $\{\tau_t\}_{t=1}^T$ that can achieve the same budget. In this section, we further investigate the impact of which task we allocate more or less of the compute budget on the final accuracy/efficiency trade-off.

We perform an experiment sweep for different combination of sparsity targets, where each $\tau_t$ is chosen from $\{0, 0.25, 0.75, 1.0\}$. Each experiment is run for two different random seeds and two different sparsity loss weights $\lambda_s$. Due to the large number of experiments, we perform the ablation experiments for shorter training runs (75% of the training epochs for each setup)

Our take-away conclusions are that **(i)** we clearly observe that some tasks require more task-specific parameters (hence a higher sparsity target) and **(ii)** this dichotomy often correlates with the per-task performance gap observed between the STL and MTL baselines, which can thus be used as a guide to set the hyperparameter values for $\tau$.

In the results of NYUD-v2 in Figure 5, we observe a clear hierarchy in terms of task importance: When looking at the points on the Pareto curve, they prefer high values of $\tau_{\text{normals}}$, followed by $\tau_{\text{segmentation}}$: In other words, these two tasks, and in particular normals prediction, requires more task-specific parameters than the depth prediction task to obtain the best MTL performance versus compute cost trade-offs.

Then, we conduct a similar analysis for the five tasks of PASCAL-Context in Figure 6. Here we see a clear split in tasks: The graph for the edges prediction and saliency task are very similar to one another and tend to prefer high $\tau$ values, i.e. more task-specific parameters, at higher compute budget. But when focusing on a lower compute budget, it is more beneficial to the overall objective for these tasks to use the shared branch. Similarly, the tasks of segmentation and human parts exhibit similar behavior under variations of $\tau$ and are more robust to using shared representations (lower values of $\tau$). Finally, the task of normals prediction (b) differ from the other four, and in particular exhibit a variance of behavior across different compute budget. In particular, when targeting the intermediate range (350B-450B FLOPs), setting higher $\tau_{\text{normals}}$ helps the overall objective.

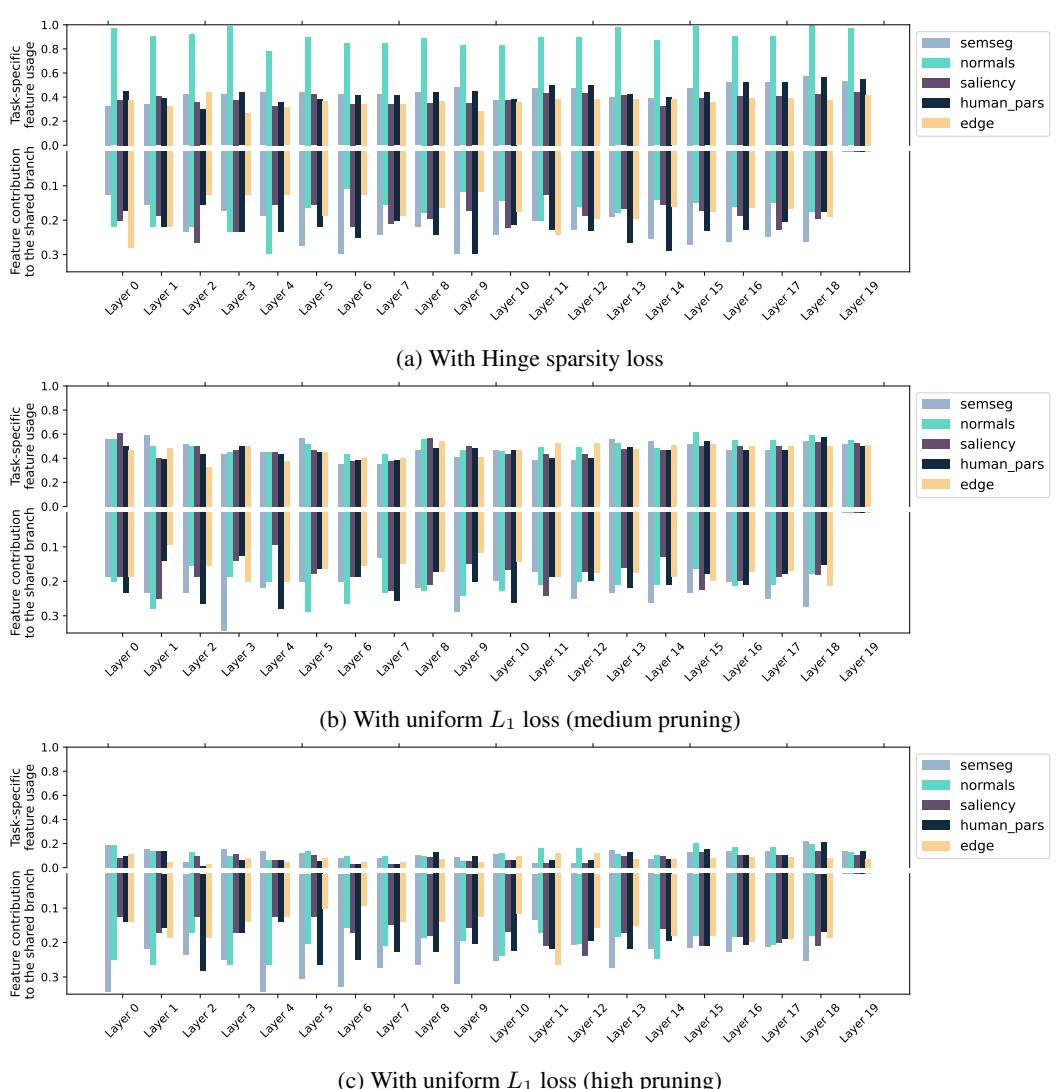

(a) With Hinge sparsity loss

(b) With uniform $L_1$ loss (medium pruning)

(c) With uniform $L_1$ loss (high pruning)

Figure 4: sharing and specialization patterns on pascal context dataset with ResNet-18 backbone.

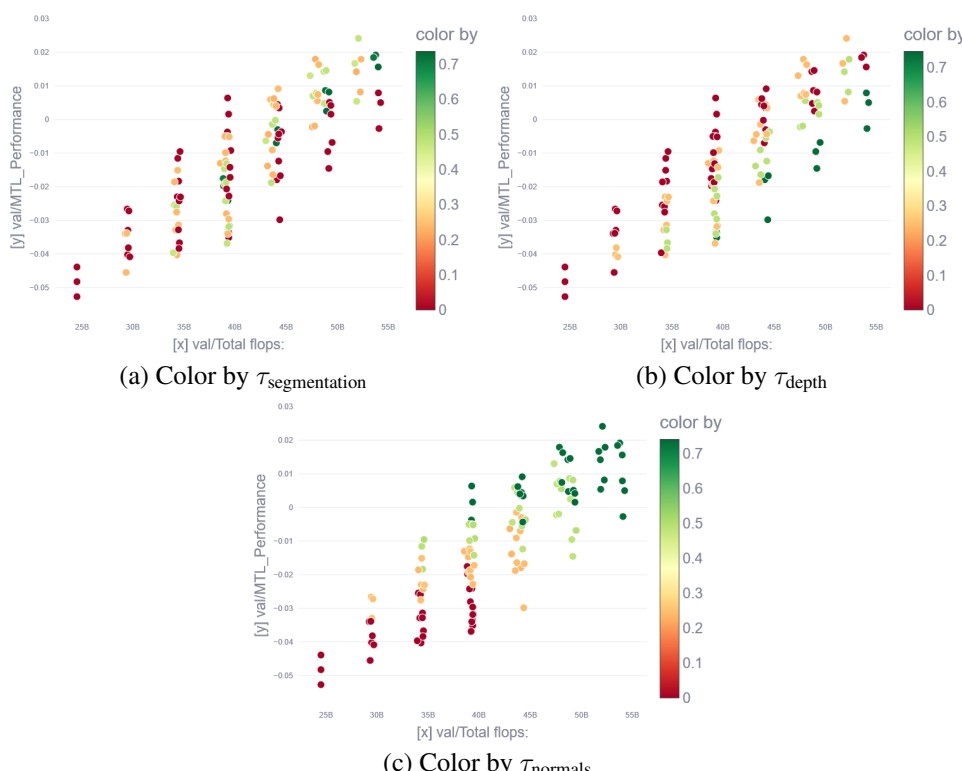

Figure 5: Sweeping over different $\{\tau_t\}$ on the NYUD-v2 experiments with HRNet-18 backbone. We plot the MTL performance $\Delta_{MTL}$ against the total number of FLOPs, then color each scatter point by the value of $\tau_t$ when the task $t$ is (a) segmentation, (b) depth and (c) normals.

## D  FORWARD-PASS PSEUDO-CODE

Algorithm 1 illustrates the steps in the forward pass of the algorithm

---

**Algorithm 1** Pseudo-code for unified representation encoder

---

Given:
- $x \in \mathbb{R}^{3 \times W \times H}$        ▷ Input image
- $T, L \in \mathbb{R}$        ▷ Number of tasks and encoder layers
- $\Psi, \Phi_t$        ▷ shared and $t$-th task-specific layer parameters
- $\beta, \alpha_t$        ▷ shared and $t$-th task-specific gating parameters

Return: $[\varphi_1^L, ..., \varphi_T^L]$        ▷ the task-specific encoder representations
$\psi^0, \varphi_1^0, ..., \varphi_T^0 \leftarrow x$        ▷ Set initial shared and task-specific features
**for** $\ell = 1$ to $L$ **do**
    **for** $t = 1$ to $T$ **do**
        $\varphi_t'^\ell \leftarrow G_t^\ell(\alpha_t^\ell) \odot \varphi_t^\ell + (1 - G_t^\ell(\alpha_t^\ell)) \odot \psi^\ell$ (2) ▷ Choose shared and task-specific features
        $\varphi_t^{\ell+1} \leftarrow F(\varphi_t'^\ell; \Phi_t^\ell)$        ▷ Compute task-specific features
    **end for**
    $\psi'^\ell = \sum_{t=1}^T \underset{t=1...T}{\text{softmax}}(\beta_t^\ell) \odot \varphi_t'^\ell$ (3)        ▷ Combine task-specific features to form shared ones
    $\psi^{\ell+1} \leftarrow F(\psi'^\ell; \Psi^\ell)$        ▷ Compute shared features
**end for**

---

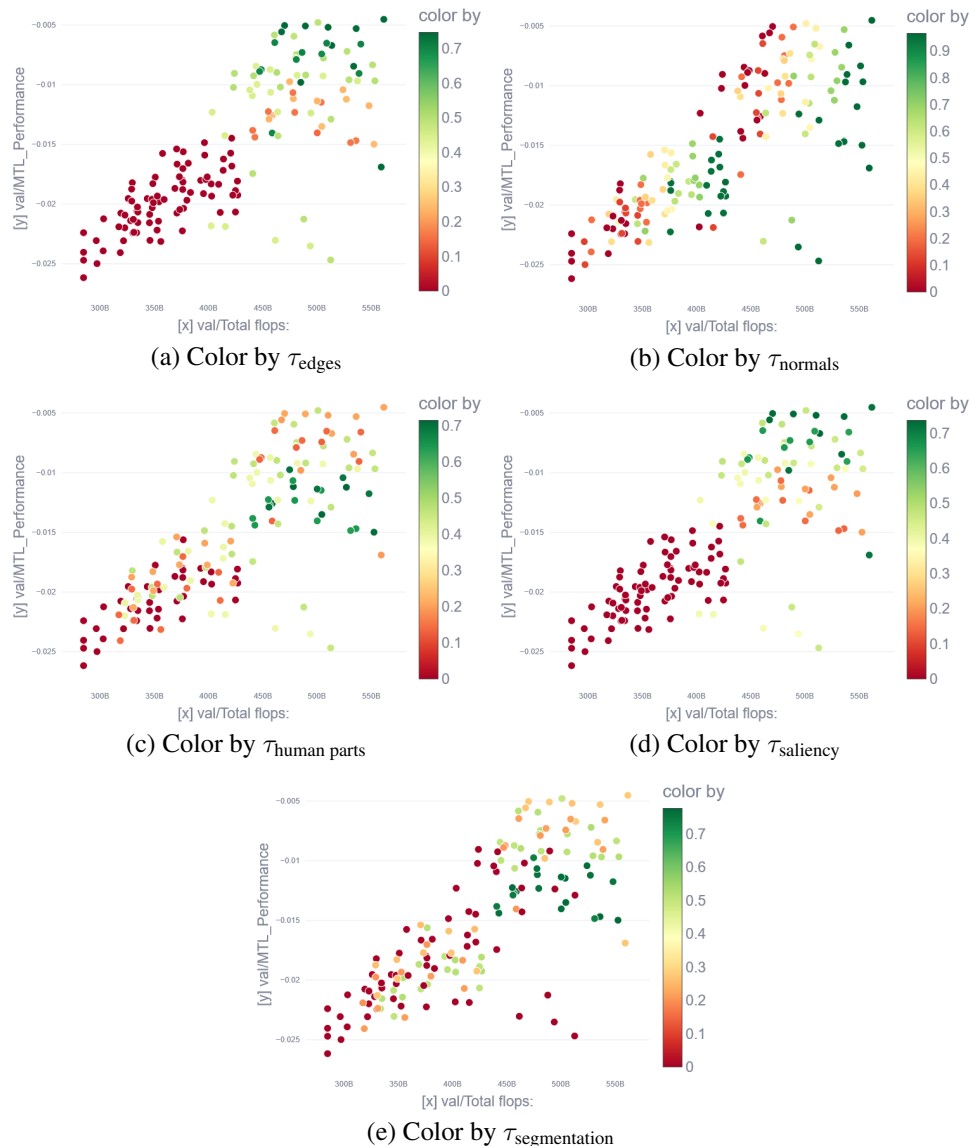

Figure 6: Sweeping over different $\{\tau_t\}$ on the PASCAL-Context. We plot the MTL performance $\Delta_{MTL}$ against the total number of FLOPs, then color each scatter point by the value of $\tau_t$ when the task $t$ is (a) edges, (b) normals (c) human parts, (d) saliency and (e) segmentation.

# E  GENERALIZATION TO VISION TRANSFORMERS

As transformers are becoming widely used in the vision literature, and to show the generality of our proposed MTL framework, we also apply GatedMTL to vision transformers: We again denote $\varphi_t^\ell \in \mathbb{R}^{N^\ell \times C^\ell}$ and $\psi^\ell \in \mathbb{R}^{N^\ell \times C^\ell}$ as the $t$-th task-specific and shared representations in layer $\ell$, where $N^\ell$ and $C^\ell$ are the number of tokens and embedding dimensions, respectively. We first apply our feature selection to the key, query and value linear projections in each self-attention block:

$$q_t^{l+1} = G_t^\ell(\alpha_{q,t}^\ell) \odot f_{q,t}^\ell(\varphi_t^\ell; \Phi_t^\ell) + (1 - G_t^\ell(\alpha_{q,t}^\ell)) \odot f_q^\ell(\psi^\ell; \Psi^\ell), \tag{7}$$

$$k_t^{l+1} = G_t^\ell(\alpha_{k,t}^\ell) \odot f_{k,t}^\ell(\varphi_t^\ell; \Phi_t^\ell) + (1 - G_t^\ell(\alpha_{k,t}^\ell)) \odot f_k^\ell(\psi^\ell; \Psi^\ell), \tag{8}$$

$$v_t^{l+1} = G_t^\ell(\alpha_{v,t}^\ell) \odot f_{v,t}^\ell(\varphi_t^\ell; \Phi_t^\ell) + (1 - G_t^\ell(\alpha_{v,t}^\ell)) \odot f_v^\ell(\psi^\ell; \Psi^\ell), \tag{9}$$

Table 9: Training time comparison of various MTL methods

| Method | Forward (ms) | Backward (ms) | Training time (h) | $\Delta_{\text{MTL}}$ |
|---|---|---|---|---|
| Standard MTL | 60 | 299 | 7.5 | -4.14 |
| MTAN | 73 | 330 | 8.5 | -1.78 |
| Cross-stitch | 132 | 454 | 12.3 | +0.14 |
| MGDA-UB | 60 | 568 | 13.2 | -1.94 |
| CAGrad | 60 | 473 | 11.1 | -2.03 |
| PCGrad | 60 | 495 | 11.6 | -2.58 |
| GatedMTL | 76 | 324 | 8.4 | -1.35 |
| GatedMTL | 102 | 376 | 10.1 | +0.12 |
| GatedMTL | 119 | 426 | 11.5 | +0.42 |

where $\alpha_{q,t}^{\ell}$, $\alpha_{k,t}^{\ell}$, $\alpha_{v,t}^{\ell}$ are the learnable gating parameters mixing the task-specific and shared projections for queries, keys and values, respectively. $f_{q,t}^{l}$, $f_{k,t}^{l}$, $f_{v,t}^{l}$ are the linear projections for query, key and value for the task $t$, while $f_{q}^{l}$, $f_{k}^{l}$, $f_{v}^{l}$ are the corresponding shared projections. Once the task-specific representations are formed, the shared embeddings for the next block are computed by a learned mixing of the task-specific feature followed by a linear projection, as described in (3). Similarly, we apply this gating mechanism to the final linear projection of the multi-head self-attention, as well as the linear layers in the feed-forward networks in-between each self-attention block.

## F TRAINING TIME COMPARISONS

While our method is mainly aiming at improving the inference cost efficiency, we also measure and compare training times between our method and the existing literature, on the PASCAL-Context Chen et al. (2014). The results are shown on Table 9. The forward and backward iterations are averaged over 1000 iterations, after 10 warmup iterations, on a single NvidiaV100 GPU, with a batch size of 4.

