# OpenReview forum: "GatedMTL: Learning to Share, Specialize, and Prune Representations for Multi-task Learning"
_ICLR.cc/2024/Conference — Submitted to ICLR 2024_

### Official Review · Reviewer_7nfe · 2023-10-26

**Soundness:** 2 fair
**Presentation:** 2 fair
**Contribution:** 2 fair
**Rating:** 5
**Confidence:** 4

**Summary:**

This paper proposes a novel multi-task learning (MTL) framework, GatedMTL, to address the fundamental challenges of task interference and computational constraints in MTL. GatedMTL learns the optimal balance between shared and specialized representations by leveraging a learnable gating mechanism to allow each task to select and combine channels from its task-specific features and a shared memory bank of features. Moreover, a regularization term is used to learn the optimal balance between allocating additional task-specific parameters and the model’s computational costs. Extensive empirical evaluations are conducted.

**Strengths:**

1. This paper proposes a novel multi-task learning (MTL) framework to address the fundamental challenges of task interference and computational constraints in MTL.
2. Extensive empirical evaluations are conducted.

**Weaknesses:**

1. The code is not provided.
2. The description of the proposed method in Section 3 and the overall framework in Figure 1 are confusing. If my understanding is correct, the proposed method is very similar to the existing MoE-base MTL methods. However, this paper does not discuss and compare with MoE-based MTL methods.
3. The proposed method uses the single-task weights for initialization, which means it needs to train $T$ single-task models before training the proposed method, and it is unfair to compare with the baselines which do not use the information from single-task models.

See the next Questions part for details.

**Questions:**

**Major Concerns**:
1. The description in Section 3 and Figure 1 are confusing. Is the encoder in Figure 1 shared among different tasks? Does $\Psi$ denote the shared encoder in Figure 1? If so, which part in Figure 1 is $\Phi_t$, and how can we obtain the shared and task-specific features at each layer? Are there $T+1$ encoders where one is $\Psi$ shared among different tasks and the others are task-specific $\Phi_t$? If so, what is the difference between the proposed GatedMTL and MoE-based MTL methods like [1, 2, 3, 4, 5]?
2. How to choose $\omega_t$ in Eq. (1)?
3. In the last paragraph of Section 4.1: "the task-specific branches are with their corresponding single-task weights". It means we need to train $T$ single-task models before training the proposed GatedMTL, which causes a huge computational cost in the training process.
4. "for a given computational budget" in the abstract and "matching the desired target computational cost" in the second contribution. What is the "given computational budget" or "desired target computational cost"? Is it $\tau_t$ in Eq. (4)? However, $\tau_t$ represents neither parameter size nor flops. Besides, although both $\lambda_s$ and $\tau_t$ can control the trade-off between performance and computational cost, the sparsity regularization cannot be guaranteed to be optimized to $0$.
5. Why not report and compare the parameter size? It is very important in multi-task learning.
6. Many recent or important baselines are missing. For example, MoE-based MTL methods like [1, 2, 3, 4, 5] and MTO approaches like [6, 7, 8].


**Minor Concerns**:
1. $\odot$ in Eqs. (2), (3), (6), (7), and (8) is not defined.
2. Next line of Eq. (3): $R$ should be $\mathbb{R}$.
3. $\beta^l$ is a learnable parameter, but it does not appear in the overall training objective Eq. (5).
4. $(\tau_t)\_{t=1}^T$ should be $\\{\tau_t\\}\_{t=1}^T$.
5. Some references appear twice, such as "Multi-task learning using uncertainty to weigh losses for scene geometry and semantics.", "End-to-end multi-task learning with attention.", "Auto-lambda: Disentangling
dynamic task relationships.", and "Attentive single-tasking of multiple tasks.".

----
[1] Modeling Task Relationships in Multi-task Learning with Multi-gate Mixture-of-Experts. KDD, 2018.

[2] Progressive Layered Extraction (PLE): A Novel Multi-Task Learning (MTL) Model for Personalized Recommendations. RecSys, 2020.

[3] DSelect-k: Differentiable Selection in the Mixture of Experts with Applications to Multi-Task Learning. NeurIPS, 2021.

[4] Deep Safe Multi-Task Learning. arXiv:2111.10601v2.

[5] MSSM: A Multiple-level Sparse Sharing Model for Efficient Multi-Task Learning. SIGIR, 2021.

[6] Multi-Task Learning as Multi-Objective Optimization. NeurIPS, 2018.

[7] Conflict-Averse Gradient Descent for Multi-task Learning. NeurIPS, 2021.

[8] Reasonable Effectiveness of Random Weighting: A Litmus Test for Multi-Task Learning. TMLR, 2022.

---

> ### Author Response · Authors · 2023-11-18
>
> We thank the reviewer for their thorough review of our paper and insightful comments.
>
> **Open-source Code Availability:** We acknowledge the critical role of open-sourcing in advancing research and reproducibility. We have initiated the process for obtaining legal approval to open-source our code. Additionally, to facilitate understanding and reproducibility in the interim, we reported all detailed hyperparameters and have included detailed pseudo-code for GatedMTL in Appendix E.
>
> **Clarification of our method:** We apologize for any confusion caused and appreciate the opportunity to clarify. The encoder in GatedMTL consists of both shared parameters $\Psi$ and task-specific parameters $\Phi_t$. The shared $\psi$ and task-specific features $\varphi_t$ are computed using their respective parameters. In Figure 1, we illustrate the architecture of a *single GatedMTL layer* within the *unified encoder*, highlighting how task-specific gates blend shared and task-specific features to form $\varphi_t^{\prime}$, serving as the input to the following task-specific layers. The purple blocks represent the features extracted from the shared branch, while the green and yellow blocks represent features extracted by task $1$ and task $t$ (we show only two task-specific branches for simplicity). For each task, a corresponding task-specific gate (e.g. yellow round block) makes a selection between using shared features and task specific features resulting in $\varphi_t^{\prime}$ shown as a blended representation with two colors. Subsequently, $\varphi_t^{\prime}$ represents the input to the next task-specific non-linear transformations with parameters $\Phi_t$. The input to the next shared layer is constructed via a linear combination of the task-specific features of all tasks, followed by a non-linear transformation parameterized by $\Psi$. There are fundamental differences between GatedMTL and MoE-based approaches which we highlight in the next point.
>
> **Comparison to MoE-based approaches:**
> A major difference between GatedMTL and MoE-based approaches is in how they operate at inference. GatedMTL is designed to solve *all tasks simultaneously* in one forward pass, a strict requirement in many real-world practical scenarios (like XR and autonomous driving applications requiring edge deployment for solving multiple tasks simultaneously). In the contrary, existing MoE-based approaches, can only solve *one task at a time*, requiring one forward pass per task. In our setting, this makes MoE approaches more expensive than conventional MTL approaches, considering the cost of the accumulated forward passes. Unlike GatedMTL which targets efficient inference for solving all tasks simultaneously, MoE-based approaches focus on improving accuracy via dynamic routing to expert modules while solving one task at a time. Therefore, MoE models are not suitable for the setup we are targeting in this work. We have made these clearer in the manuscript and highlighted the key differences in the related work section.
>
> **Using single-task weights and implications on training cost:** Our work primarily aims to enhance inference efficiency, vital for deploying multi-task learning (MTL) models in edge computing environments. In numerous real-world applications, the gains in efficient inference over time, especially when scaled across millions of edge devices, can significantly offset the initial higher training costs. While we acknowledge the importance of training efficiency and have discussed GatedMTL’s training cost limitations in our manuscript, it’s important to highlight that competing methods like MTO approaches (PCGrad [1], CAGrad [2], MGDA [3], etc.) also bear considerable training costs, requiring multiple backward passes during training. Our GatedMTL significantly surpasses these in performance, demonstrating its advantage for practical applications that demand efficient inference on the edge.
>
> In addition, in both MTL research and practice, establishing single-task baselines is a standard practice to gauge the relative performance of MTL models. Therefore, having access to single task models is not unreasonable from a practical perspective. In addition, pre-trained single task models for many applications are abundantly available which may obviate the need to separately train STL models.

---

> ### Author Response · Authors · 2023-11-18
>
> **Comparison to recent baselines and parameter size:**
> In response to the reviewers feedback regarding the selection of baselines, we have extensively expanded our experiments to include the suggested state-of-the-art methods, i.e. PCGrad [1], CAGrad [2], MGDA [3], and RDW [4]. The results are presented in the table below. As can be seen, GatedMTL substantially outperforms all competing baselines across both $\Delta_{MTL}$ and the newly added Mean Rank metric (suggested by R1). We additionally report the parameter counts of the models as suggested by the reviewer. Our method compares favorably against Cross-stitch and MTAN in terms of number of parameters and performance. Please note that as discussed above, our comparisons do not include MoE-based approaches as they are not suitable for our MTL setup for solving all tasks simultaneously.
>
>
> Updated performance comparison on NYUD-v2 using HRNet-18 backbone. The new baselines are shown in bold.
> |Model|Semseg|Depth|Normals|$\Delta_\text{MTL}$(\%)|Flops(G)|Param(M)|MR|
> |-|-|-|-|-|-|-|-|
> |STL|41.70|0.582|**18.89**|0|65.1|28.9|8.0|
> |MTL(Uni.)|41.83|0.582|22.84|-6.86|24.5|9.8|11.0|
> |DWA|41.86|0.580|22.61|-6.29|24.5|9.8|8.7|
> |Uncertainty|41.49|0.575|22.27|-5.73|24.5|9.8|8.3|
> |Auto-$\lambda$|42.71|0.577|22.87|-5.92|24.5|9.8|7.7|
> |**RDW**|42.10|0.593|23.29|-8.09|24.5|9.8|11.7|
> |**PCGrad**|41.75|0.581|22.73|-6.70|24.5|9.8|10.3|
> |**MGDA**|41.23|0.625|21.07|-6.68|24.5|9.8|11.3|
> |**CAGrad**|42.31|0.580|22.79|-6.28|24.5|9.8|8.7|
> |GatedMTL|**43.58**|**0.557**|19.32|**+2.18**|43.2|18.8|**1.3**|
> |GatedMTL|42.93|0.5613|19.72|+0.70|38.4|16.5|2.3|
> |GatedMTL|42.43|0.5671|20.01|-0.54|36.0|15.4|4.0|
> |GatedMTL|42.85|0.5778|21.01|-2.58|33.0|13.7|5.0|
> |GatedMTL|42.35|0.5755|21.70|-4.06|29.1|11.9|6.0|
>
> -----
> Updated performance comparison on NYUD-v2 using ResNet-50 backbone.
> |Model|Semseg|Depth|Normals|$\Delta_\text{MTL}$(\%)|Flops(G)|Param(M)|MR|
> |-|-|-|-|-|-|-|-|
> |STL|43.20|0.599|**19.42**|0|1149|118.9|9.0|
> |MTL(Uni.)|43.39|0.586|21.70|-3.04|683|71.9|9.7|
> |DWA|43.60|0.593|21.64|-3.16|683|71.9|9.7|
> |Uncertainty|43.47|0.594|21.42|-2.95|683|71.9|10.0|
> |Auto-$\lambda$|43.57|0.588|21.75|-3.10|683|71.9|10.0|
> |**MGDA**|42.56|0.586|21.76|-3.83|683|71.9|11.3|
> |**RDW**|43.49|0.587|21.54|-2.74|683|71.9|8.3|
> |**PCGrad**|43.74|0.588|21.55|-2.66|683|71.9|7.3|
> |**CAGrad**|43.57|0.583|21.55|-2.49|683|71.9|7.0|
> |MTAN|**44.92**|0.585|21.14|-0.84|683|92.4|4.0|
> |Cross-stitch|44.19|0.577|19.62|+1.66|1151|119.0|2.3|
> |GatedMTL|44.15|**0.573**|19.49|**+2.08**|916|95.4|**2.0**|
> |GatedMTL|43.70|0.578|19.65|+1.16|892|92.4|4.0|
> |GatedMTL|42.99|0.589|19.93|-0.48|798|83.3|9.7|
>
> **The choice of $\omega_t$:** The hyper-parameter $\omega_t$ denotes the scalarization weights. We use the weights suggested in prior work but also report numbers of uniform scalarization. For NYUD-v2, we use uniform scalarization as suggested in [5,6], and for PASCAL-Context, we similarly use the weights suggested in the paper [5,6].
>
> **Target computational cost:** We agree with the reviewer and reformulated the wording of the claim in the abstract and additionally mentioned this as a limitation of our method. As correctly described by the reviewer, although both $\lambda_s$ and $\tau_t$ can control the trade-off between performance and computational cost and closely result in the desired FLOPs count, we still cannot guarantee landing on a particular target computational cost.
>
> **Minor comments:** We thank the reviewer for their detailed and meticulous comments. We fixed all of them directly in the manuscript.
>
>
>
> References:
>
> [1] Gradient Surgery for Multi-Task Learning. NeurIPS, 2020.
>
> [2] Conflict-Averse Gradient Descent for Multi-task Learning. NeurIPS, 2021.
>
> [3] Multi-Task Learning as Multi-Objective Optimization. NeurIPS, 2018.
>
> [4] Reasonable Effectiveness of Random Weighting: A Litmus Test for Multi-Task Learning. TMLR, 2022.
>
> [5] Attentive Single-Tasking of Multiple Tasks, CVPR, 2019.
>
> [6] Multi-task learning for dense prediction tasks: A survey, PAMI, 2021.

---

> ### Comment · Reviewer_7nfe · 2023-11-23
>
> Thanks to the authors for the rebuttal. I still have some concerns about my initial comments.
>
> 1. I still do not understand the architecture of GateMTL in Figure 1. Are there $T+1$ encoders where one is shared among different tasks and the remaining $T$ are task-specific? If so, why the parameter size of GateMTL is smaller than STL? If not, how do we obtain each layer's shared and task-specific features?
>
> 2. Figure 1 in [2] summarizes different MTL architectures. The blue block means the shared block and the red and green ones are task-specific. (1) The architecture of PLE is similar to GateMTL. (2) Why MoE-based methods can only solve a task at a time?
>
> 3. I appreciate that this paper aims to enhance inference efficiency, but is it related to initializing task-specific branches with their corresponding single-task weights? It is unfair to compare with the baselines that do not use the information from single-task models. I suggest training GateMTL without single-task weights initialization for a fair comparison.
>
> 4. The method name in [8] is RLW rather than RDW.

---

> > ### Author Response · Authors · 2023-11-23
> >
> > 1) Our GateMTL model **initially** comprises one shared encoder and $T$ task-specific encoders. As the training progresses, the gates at each layer adaptively learn to select between shared and task-specific parameters for every task. By leveraging the sparsity loss, the model converges towards utilizing a substantial portion of the shared representations, supplemented by a minimal yet crucial set of task-specific features that are most beneficial for each task. Essentially, the model optimizes its architecture through the gating mechanism, **learning to prune** itself during training. This pruning is what enables GateMTL to achieve significantly lower number of parameters and computational costs compared to separate single-task (STL) models, despite starting with a similar parameter size.
> >
> >
> > 2) There are several fundamental differences between MoE architectures such as PLE and GatedMTL.
> > - Firstly, as can be seen in Equation (2) and Figure 4 of the PLE paper, the gates are *input dependent* and combine the chosen experts together. This makes PLE a **dynamic architecture at inference**, requiring to load the relevant expert weights (out of many) into memory at each layer. As can be seen in the bottom right side of Figure 1 of our manuscript, the gates in GatedMTL are task-dependent (not input dependent), and are there only at training time. At inference, *the gates are completely removed* from the architecture, and **the parameters that were not chosen are permanently removed from the model**. This makes GatedMTL far more parameter efficient, and lightweight at inference. This information has been mentioned in the manuscript in section 3.1.
> > - Secondly, PLE utilizes a collection of experts for each task (see $E_{A1}$, $E_{A2}$, $E_{A3}$, etc. in Figure 4 of [2]), and selects among them dynamically in a input-dependent fashion. GatedMTL does not have any experts to choose from. During training rather than selecting among experts, GatedMTL chooses between shared and task-specific parameters (convolutional kernels in case of CNNs), **where the selection of shared parameters, translates to fully pruning the non-selected take-specific features from the architecture**. As can be seen in our Figure 1, bottom right, GatedMTL has non-symmetric architecture at inference, which has no equivalence in an MoE-based architecture.
> >
> > - Finally, MoE's are not designed to solve multiple-tasks simultaneously. This is primarily because the routing of the input inside the architecture for each task is completely different because each task at each layer uses a different subset of the experts and obtains independent feature maps. In an MoE architecture with dynamic expert selection, a single forward path has the equivalent cost of an STL model. Running all tasks simultaneously would translate to running T independent STL models with an extreme cost which is not suitable for many practical scenarios. For example, consider the state-of-the-art multi-lingual translation MoE model designed for translating between any two languages. Such a model isn’t intended to translate one language to all others simultaneously, which isn’t a primary concern in its application domain.
> >
> > We will cite PLE [2] in our manuscript and highlight these differences accordingly.
> >
> >
> >
> > 3) We disagree with the assertion that utilizing single-task model (STL) makes it unfair to compare against methods that neglect using them. Our position is that **the inability of existing MTL solutions to effectively incorporate STL weights** mainly because of inherent design limitations, represents a notable limitation in the field. **This should not, however, be a deterrent to the development and adoption of new approaches that can successfully leverage STL weights, as we propose in our work.**
> >
> > We consider our method’s ability to harness STL weights for enhancing the performance of MTL models during inference *as a strength and a novel aspect of our research*. Far from being a questionable practice, employing STL data in this manner aligns with practical machine learning principles and optimizes the use of available resources.
> >
> > In addition, in numerous real-world applications, the gains in efficient inference over time, especially when scaled across millions of edge devices, can significantly offset the additional requirement of collecting STL weights. We would like to emphasize that single task pre-trained models for tasks such as semantic segmentation, depth estimation, etc. are abundantly available for a multitude of model architectures that could be plugged-in off-the-shelf.
> >
> > However, to ensure full transparency with readers, we have acknowledged the necessity of collecting STL weights in the limitations section of our manuscript to clearly inform readers of any potential trade-offs involved in our approach.
> >
> >
> >
> > 4) Thanks for pointing this out. We will fix it in the manuscript.

---

> > > ### Comment · Reviewer_7nfe · 2023-11-23
> > >
> > > Thanks to the authors for further feedback, which addresses parts of my initial concerns.
> > >
> > > 1. I think Figure 1 should be reorganized: it should clearly show $T+1$ encoders in the initial training and some blocks are pruned in the training process.
> > >
> > > 2. I approve of the discussion between PLE and GateMTL. How about [4] in my initial review? The gates in [4] are task-dependent (not input-dependent), thus [4] can solve multiple tasks with a single forward process in the inference. It seems the difference between [4] and GateMTL is that a continuous gate is used in [4] while a discrete one is used in GateMTL.
> > >
> > > 3. I partially agree with your point of view. For a fair comparison and a comprehensive understanding of GateMTL, it is better to provide the results for GateMTL without single-task initialization. Also, it is better to compare with MoE-based methods (like [1, 2, 3, 4, 5] in my initial review) with and without single-task initialization.

---

> ### Author Response · Authors · 2023-11-23
>
> We thank the reviewer for this discussion.
>
> **Comparison to [4]:** We will improve the visualization in Figure 1. It appears there are some misunderstandings about how GatedMTL works. In [4], the *DSMTL-IL* variant is extremely expensive (all encoders are executed) and incomparable to GatedMTL both methodologically and computationally. The DSMTL-AL, which focuses on saving compute at inference, has $T$ private encoders for each specific task. The main focus is how to *learn to branch out from the shared encoder* to the task-specific encoder. In other words, part of the trunk is shared among tasks, and tasks decide to exit the shared branch at some point and switch to their task-specific encoder blocks. We find this an interesting idea and will discuss it in the paper, but DSMTL-AL is not an MoE architecture. DSMTL is fundamentally different from GatedMTL. GatedMTL does not gate blocks but rather individual channels/filters making it far more flexible and fine-grained. Our work does not focus on when to branch out from a shared branch, but rather to decide how to effectively choose from every individual shared-parameter and task-specific ones, resulting in a more expressive architecture. Unfortunately, the paper [4] does not provide inference timing or source code which makes comparison difficult.
>
> **Regarding the comparison with MoE-based** methods starting from single-task models, two primary challenges hinder their suitability in our multi-task learning (MTL) context. First, as we mentioned before, the MoE framework, by its nature, requires running tasks sequentially, leading to a cumulative inference cost equivalent to executing $T$ single tasks. This approach contradicts the efficiency principles of GatedMTL and limits the relevance of a direct comparison.
>
> Furthermore, the initialization of single-task weights into MoE architectures presents unresolved complexities. Consider an MoE with 16 experts at each layer solving three tasks, where the router dynamically selects from these experts. How is it possible to load single task weights inside these 16 experts in an MoE architecture? Another challenge arrises when we match the number of experts and number of tasks. A fundamental challenge in MoEs is jointly learning the router and experts. The router needs to direct samples to the most appropriate experts, but these experts only become proficient if they receive the correct samples by the router. Initializing experts with single-task weights in this scenario can lead the router to **trivial solutions** selecting the corresponding expert for each task (mode collapse), thereby ignoring other experts. This pattern contradicts the intended learning dynamics of MoEs and undermines the model’s overall adaptability and learning potential.

---

### Official Review · Reviewer_DXQP · 2023-10-30

**Soundness:** 2 fair
**Presentation:** 3 good
**Contribution:** 2 fair
**Rating:** 5
**Confidence:** 4

**Summary:**

The paper proposes a GatedMTL framework for MTL.  GatedMTL aims to address the fundamental challenges of task interference and computational constraints in MTL. Specifically, a learnable gating mechanism is used to select and combine channels from its task-specific features and a shared memory bank of features. In addition, the gates are regularized to learn the optimal balance between allocating additional task-specific parameters and the model’s computational costs. The proposed method is evaluated on datasets and the experiment results also achieve comparable performance. However, the contribution of this GatedMTL seems marginal and the results are not very strong.

**Strengths:**

1) The proposed GatedMTL method to assign features to either a task-specific or shared branch, until reaching an adjustable target computational budget.
2) Experiment results demonstrate competitive performance.
3) Easy to understand.

**Weaknesses:**

1) The core idea of this paper is to find a parameter to control the ratio of task-specific features to task-shared features. The motivation of the gating design for MTL is not clear. The gating mechanism is not a new story in MTL.
2) The gating module to balance task-specific features and the shared features in the decoder seems a bit more reasonable. Since the encoder is responsible for encoding out the shared features across all tasks, it doesn't seem to make sense to split out the task-specific features in the encoder.
3) The proposed gating mechanism seems similar to a simplified variant of smooth Gating in DSelect-k[R1]. It is not possible to observe from Eqs. 2 and 6 that there is a point of novelty in the gating of this paper.
4) The authors are encouraged to show comparisons of feature changes before and after the addition of gating through visualization. In addition, how to show task-specific features and shared features. Can these two features be displayed through visualization?
5) The results in Table 1 were confusing to the reviewers, who could not see directly from the table how the five GatedMTLs are differentiated. The other tables have the same confusion.
6) Why are the results for Auto-λ not shown in Tables 3 and 4?
[R1] DSelect-k: Differentiable Selection in the Mixture of Experts with Applications to Multi-Task Learning, NeurIPS, 2021.

**Questions:**

1) The gating module to balance task-specific features and the shared features in the decoder seems a bit more reasonable. Since the encoder is responsible for encoding out the shared features across all tasks, it doesn't seem to make sense to split out the task-specific features in the encoder. Have the authors considered this?
2) Minor error:
$\Delta_{MTL}$ and $\Delta$ denote the same metric. The authors are encouraged to keep them consistent.

---

> ### Author Response · Authors · 2023-11-18
>
> We thank the reviewer for their thorough review of our paper and insightful comments.
>
> **The motivation of the gating design for MTL:** We would like to highlight the core motivation for the design of our GatedMTL model. The core concept of our paper addresses a significant challenge in MTL: *task interference* [1]. This phenomenon is particularly evident in models with limited representation capacity. In such models, while certain features are universally beneficial across all tasks, task-specific features often compete for the remaining capacity. This interference can lead to dominance of certain tasks over others during training, influenced by the dynamics of training and the magnitudes of loss functions.
>
> Our primary innovation lies in recognizing that a rigid or pre-defined allocation of feature capacity between task-specific and shared features is sub-optimal. Such static partitioning fails to adapt to the varying needs of different tasks and layers within the network. To overcome this, we propose a flexible gating mechanism. This mechanism is not just a tool for balancing feature ratios; its fundamental purpose is to learn the optimal architecture dynamically, adapting the degree of feature sharing and task-specific specialization at each layer for each task. This approach ensures that each task has access to the necessary feature capacity without the interference of gradients from other tasks, while at the same time allowing for maximizing the sharing of relevant features.
>
> It’s important to clarify that our contribution is not merely the gating mechanism itself, which, as the reviewer correctly notes, is not a novel concept in MTL. The novelty of our work lies in the application of this mechanism to dynamically learn and adjust the architecture for optimal feature sharing and specialization. This is a significant departure from traditional MTL models, where such parameters are usually fixed and do not adapt to the specific requirements of each task, while being highly efficient at the inference in solving all tasks at once.
>
> **Where to gate, encoder or decoder?** We thank the reviewer for bringing up this interesting discussion. We absolutely agree that in typical MTL setups, the encoder is responsible for encoding out the *shared features across all tasks*. As highlighted in the above discussion, task interference is inherently the problem that arises with sharing representations across several tasks and the encoder is a natural choice because it is fully shared among all tasks. We would like to provide three additional major reasons for why we chose to resolve task interference within the encoder rather than the decoder:
>
> 1. **Computational efficiency and specialization of decoders:** Our design is grounded in the conventional MTL architecture, which typically comprises a large shared backbone and multiple task-specific decoders, as depicted in Figure 1 (left). In such architectures, the majority of computational resources are consumed by the encoder (backbone). For instance, in the context of NYUD-v2 with three tasks, the cumulative computational load of the HRNet18 decoders constitutes merely 17\% of the total. Given that the decoders are designed to be highly specialized and independent for each task to optimize performance, it is both computationally efficient and logical to address task interference predominantly in the backbone.
>
> 2. **Diversity in task decoding requirements:** In MTL, particularly in complex applications like autonomous driving, tasks can vary greatly in their nature, such as combining dense prediction tasks, object detection tasks, and classification tasks such as weather prediction. In such scenarios, decoders might possess distinctly different architectures or layers. Forcing feature sharing at the decoder level could be detrimental, as it disregards the unique requirements and structures of each task’s decoder.
>
> 3. **Encoder feature quality and decoder success:** The quality of features extracted by the encoder is pivotal for the success of the specialized decoders. Significant task interference at the encoder stage can lead to degraded feature quality, resulting in irreversible information loss and sub-optimal input for the decoders. Ensuring high-quality feature extraction by the encoder is therefore crucial for the effective functioning of the task-specific decoders.
>
> Finally, our empirical evaluations strongly demonstrate the effectiveness of the encoder-based approach to mitigating task-interference. We leave the investigation of decoder-based approaches to future work and discuss that in the manuscript.

---

> > ### Author Response · Authors · 2023-11-18
> >
> > **Distinction from DSelect-k:** Firstly, it is important to clarify that we do not claim innovation in the design of our gating modules. Our gates are simple binary units that make a binary decision between task-specific features and shared features, and they are trained using the well-established straight-through estimation (STE) [2] by Bengio, et al. This is fundamentally different from the DSelect-k mechanism, which is tailored for MoE architectures to select a limited number of experts (at most $k$ out of $n$) in a continuous and differentiable manner. Our method does not involve a Top-K selection process, thereby making the comparison to DSelect-k less applicable.
> >
> > Furthermore, there are significant differences between GatedMTL and MoE-based methods. MoE architectures handle one task at a time, whereas our GatedMTL is designed to solve all tasks simultaneously. This simultaneous task-solving capability is crucial for a wide range of real-world applications, setting GatedMTL apart from MoE-based approaches.
> >
> > **Visualizing task-specific and shared features:** We thank the reviewer for this insightful suggestion regarding the visualization of feature changes before and after the implementation of gating, as well as the distinction between task-specific and shared features in our model. We acknowledge the potential value of such visualizations in enhancing the understanding of our model’s behavior and the effectiveness of the gating mechanism. Currently, our paper focuses on quantitatively demonstrating the improvements of our proposed GatedMTL model in terms of accuracy, efficiency, and reduced task interference. Visualizing the feature space in deep neural networks, especially in the context of MTL, is a complex and non-trivial task. It involves intricacies such as high-dimensional data representation, the dynamic nature of task-specific and shared features, and the interpretability of such visualizations. While we recognize the value of this approach, we consider your suggestion as a valuable direction for our future research.
> >
> > **Clarification on multiplicity of GatedMTL entries in results**: Among the advantages of our proposed method, the user can control the operating point in the accuracy-efficiency trade-off. The multiple reported GatedMTL points in Tables 1-6 and 8 as well as the curves in Figure 2 are obtained by setting different sparsity loss ($\mathcal{L}_\text{sparsity}$) coefficients ($\lambda_s$). We thank the reviewer for bringing it up; We have further clarified this in the text of the manuscript and captions of the tables.
> >
> > **Missing results for Auto-$\lambda$ in Tables 3 and 4:** Despite extensive hyperparameter search for Auto-$\lambda$, the experiments using Transformer backbones diverged with exploding losses. We also observed similar convergence issues when using Auto-$\lambda$ on a larger number of tasks such as Pascal-Context. Unfortunately, because of divergent training, the results were not reasonable to be reported.
> >
> > **$\Delta_{MTL}$ and $\Delta$ notations:** We thank the reviewer for pointing this out. We fixed all the inconsistencies.
> >
> > ---
> >
> > References:
> >
> > [1] Xiangyun Zhao, et al., A Modulation Module for Multi-task Learning with Applications in Image Retrieval, ECCV 2018.
> >
> > [2] Yoshua Bengio et al, Estimating or propagating gradients through stochastic neurons for conditional computation, arXiv preprint arXiv:1308.3432 (2013).

---

> > ### Comment · Reviewer_DXQP · 2023-11-23
> > **Thanks to the authors for the responses. This  response addresses some of my concerns.**
> >
> > 1. Whether the location of the gating mechanism should be placed in the encoder or decoder is indispensable for ablation experiments. In the decoder, the use of gating can be set to shared gating, which reduces the number of parameters and the amount of computation.
> > 2. The gating mechanism can even be considered to be placed in both encoder and decoder.

---

> > > ### Author Response · Authors · 2023-11-23
> > >
> > > We completely agree with the reviewer regarding the feasibility and the added value of studying the decoder-based gating. We have added this discussion in the limitation section as our future work. Thank you for this suggestion.

---

### Official Review · Reviewer_PvDp · 2023-11-01

**Soundness:** 3 good
**Presentation:** 2 fair
**Contribution:** 2 fair
**Rating:** 5
**Confidence:** 3

**Summary:**

The manuscript proposes a new Multi-Task Learning (MTL) framework called *GatedMTL* that learns the optimal balance between shared and task-specific representations for a given computational budget. It uses a gating mechanism to learn a combination of shared and task-specific features for each task in each layer. Unused features and weights are pruned during inference to improve sparsity and efficiency. The framework generalizes to convolutional backbone and transformer-based backbone. Experiments on CelebA, NYUD-v2, and PASCAL-Context datasets demonstrate the proposed method maintains a favorable balance between compute costs and multi-task performance across computational budgets.

**Strengths:**

*Originality*: This work introduces a multi-head gating mechanism into feature transformation, solving the challenge of multi-task learning with an emphasis on computational efficiency.

*Quality*: The experiments are extensive.

*Clarity*: The paper is written clearly, and the figures are easy to understand.

*Significance*: The problem that this work attempts to address is important. Given the computational budget, the performance improvement is obvious.

**Weaknesses:**

W1: No source code is provided. Although the experimental setup is detailed and the results are extensive, it is still necessary to provide the code for reference and reproducibility checking.

W2: Since a shared feature branch acts like a memory bank where task-specific features can communicate, a task-specific gate still learns features from other tasks, which can cause task interference.

W3: The reported performance in each table is based on a single run. The standard deviation based on multiple random runs is highly encouraged to be provided.

**Questions:**

Q1: What is the purpose of the "convolution block" in forming the shared feature map of the next layer (line 1, page 4)?

Q2: A more detailed description of the changes made to the backbone is needed for the implementation of the gated MLT layer.

---

> ### Author Response · Authors · 2023-11-18
>
> We thank the reviewer for their thorough review of our paper and insightful comments.
>
> **Open-source Code Availability:** We acknowledge the critical role of open-sourcing in advancing research and reproducibility. We have initiated the process for obtaining legal approval to open-source our code. Additionally, to facilitate understanding and reproducibility in the interim, we reported all detailed hyperparameters and have included detailed pseudo-code for GatedMTL in Appendix E.
>
> **Interference impact on gating:** This is indeed a very interesting point raised by the reviewer. While our gating mechanism does indeed select between the shared and task-specific features, potentially influenced by task interference, it also serves as a mechanism to mitigate the negative impact of task-interference by avoiding the selection of representations with severe task interference issue. This is also supported by the results in Table 6 (first row) showing that in the absence of a sparsity objective, the gates inherently and strongly favor selecting task-specific over shared features. Additionally, our empirical findings show that in practice even with strong sparsity regularization, the gates can cope with task-interference by selecting the least interfering shared features, suggesting that task-interference at this meta-level might be less of a concern for the gates.
>
> **Multiple runs with random seeds:** All reported results for the baselines in our study were averaged across *three random seeds*. We are sorry for the oversight of not mentioning this in the original manuscript and thank the reviewer for notifying us. We have now included this information in Section 4.1. We additionally report the standard deviation for the runs in the appendix. For GatedMTL runs, we inherently had numerous results for the Pareto optimal performance vs. FLOPs curve. But based on the reviewer's suggestion we are providing the GatedMTL results for three random seeds as well. Please note that the experiments are running and in the Table below we provide the mean+std for the completed runs and will update the final manuscript with the full set of results. Note that in the Table below we additionally include results for 4 new baselines suggested by wdHJ and 7nfe.
>
> |Model|Semseg|Depth|Normals|$\Delta_\text{MTL}$(\%)|Flops(G)|MR|
> |-|-|-|-|-|-|-|
> |STL|41.70|0.582|**18.89**|0$\pm$0.12|65.1|8.0|
> |MTL(Uni.)|41.83|0.582|22.84|-6.86$\pm$0.76|24.5|11.0|
> |DWA|41.86|0.580|22.61|-6.29$\pm$0.95|24.5|8.7|
> |Uncertainty|41.49|0.575|22.27|-5.73$\pm$0.35|24.5|8.3|
> |Auto-$\lambda$|42.71|0.577|22.87|-5.92$\pm$0.47|24.5|8.0|
> |**RDW**|42.10|0.593|23.29|-8.09$\pm$1.11|24.5|11.7|
> |**PCGrad**|41.75|0.581|22.73|-6.70$\pm$0.99|24.5|10.3|
> |**CAGrad**|42.31|0.580|22.79|-6.28$\pm$0.90|24.5|8.7|
> |**MGDA**|41.23|0.625|21.07|-6.68$\pm$0.67|24.5|11.3|
> |GatedMTL|**43.58**|**0.559**|19.32|**+2.06$\pm$0.13**|43.2|**1.3**|
> |GatedMTL|42.95|0.562|19.73|+0.68$\pm$0.09|38.3|2.3|
> |GatedMTL|42.36|0.564|20.04|-0.55$\pm$0.17|36.0|4.0|
> |GatedMTL|42.73|0.575|21.01|-2.55$\pm$0.11|33.1|4.0|
> |GatedMTL|42.35|0.575|21.70|-4.07$\pm$0.38|29.2|5.7|
>
>
> |Model|Semseg|Depth|Normals|$\Delta_\text{MTL}$(\%)|Flops(G)|MR|
> |-|-|-|-|-|-|-|
> |STL|43.20|0.599|**19.42**|0$\pm$0.11|1149|9.0|
> |MTL(Uni.)|43.39|0.586|21.70|-3.04$\pm$0.79|683|9.7|
> |DWA|43.60|0.593|21.64|-3.16$\pm$0.39|683|9.7|
> |Uncertainty|43.47|0.594|21.42|-2.95$\pm$0.40|683|10.0|
> |Auto-$\lambda$|43.57|0.588|21.75|-3.10$\pm$0.39|683|10.0|
> |**RDW**|43.49|0.587|21.54|-2.74$\pm$0.09|683|8.3|
> |**PCGrad**|43.74|0.588|21.55|-2.66$\pm$0.15|683|7.3|
> |**CAGrad**|43.57|0.583|21.55|-2.49$\pm$0.11|683|7.0|
> |**MGDA**|42.56|0.586|21.76|-3.83$\pm$0.17|683|11.3|
> |MTAN|**44.92**|0.585|21.14|-0.84$\pm$0.32|683|4.0|
> |Cross-stitch|44.19|0.577|19.62|+1.66$\pm$0.09|1151|2.7|
> |GatedMTL|44.38|**0.576**|19.50|**+2.04$\pm$0.07**|916|**1.7**|
> |GatedMTL|43.63|0.577|19.66|+1.16$\pm$0.10|892|3.7|
> |GatedMTL|43.05|0.589|19.95|-0.50$\pm$0.05|794|9.7|

---

> > ### Author Response · Authors · 2023-11-18
> >
> > **The purpose of the convolutional block in the shared branch:** The convolutional layers are required to transform the representations coming from task-specific branches into a new shared representation. This ensures that the shared representations gradually evolve into more abstract representations required for solving the tasks and maintains a balance between the abstraction level of task-specific and shared representations avoiding a systematic bias toward selecting the task-specific features.
> >
> > **A more detailed description of the gated MTL layer:** We have included the pseudo-code for the implementation of the forward pass of the GatedMTL encoder layers in appendix E:
> >
> > ---
> > **Given**:
> > * $x \in \mathbb{R}^{3 \times W \times H}$    &emsp;&emsp;&emsp;&emsp;&emsp; $\rhd$ Input image
> > * $T, L \in \mathbb{R}$   &emsp;&emsp;&emsp;&emsp;&emsp; &emsp; &ensp;$\rhd$ Number of tasks and encoder layers
> > * $\Psi$, $\Phi_t$ &emsp;&emsp;&emsp;&emsp;&emsp;&emsp;&emsp;&emsp;&ensp;  $\rhd$ Shared and $t$-th task-specific layer parameters
> > * $\beta$, $\alpha_t$    &emsp;&emsp;&emsp;&emsp;&emsp;&emsp;&emsp;&emsp;&emsp;$\rhd$ Shared and $t$-th task-specific gating parameters
> >
> > **Return**: [$\varphi_1^L,..., \varphi_T^L$]  &emsp;&emsp;&emsp;$\rhd$ The task-specific encoder representations
> >
> > $\psi^0, \varphi_1^0,..., \varphi_T^0  \gets x$ &emsp;&emsp;&emsp;&ensp; $\rhd$ Set initial shared and task-specific features
> >
> > **For** $\ell=1$ to $L$ **do**
> >
> > &emsp;&emsp; **For** $t=1$ to $T$ **do**
> >
> > &emsp;&emsp;&emsp;&emsp; $\varphi^{\prime \ell}_t \gets G_t^\ell (\alpha_t^\ell) \odot \varphi_t^\ell + (1 - G_t^\ell (\alpha_t^\ell)) \odot \psi^\ell$   &emsp;&emsp;  $\rhd$ Choose among shared and task-specific features
> >
> > &emsp;&emsp;&emsp;&emsp; $\varphi_t^{\ell+1} \gets F(\varphi_t^{\prime \ell}; \Phi_t^\ell)$  &emsp;&emsp;&emsp;&emsp; &emsp;&emsp;&emsp;&emsp;&emsp;&emsp;&emsp;&emsp; $\rhd$ Compute new task-specific features
> >
> > &emsp;&emsp; $\psi^{\prime \ell} = \sum_{t=1}^{T} \underset{t=1 \dots T}{\text{softmax}}(\beta_t^\ell) \odot \varphi_t^{\prime \ell}$  &emsp;&emsp;&emsp;&emsp;&emsp;&emsp;&emsp;&emsp;&emsp;$\rhd$ Combine task-specific features to form shared representations
> >
> > &emsp;&emsp; $\psi^{\ell+1} \gets F(\psi^{\prime \ell}; \Psi^\ell)$ &emsp;&emsp;&emsp;&emsp;&emsp;&emsp;&emsp;&emsp;&emsp;&emsp;&emsp;&emsp;&emsp;&emsp;&ensp;$\rhd$ Compute new shared features
> >
> > ---
> >
> > We trust that we have addressed all the comments raised by the reviewer and are happy to further clarify any points as needed.

---

### Official Review · Reviewer_wdHJ · 2023-11-01

**Soundness:** 3 good
**Presentation:** 3 good
**Contribution:** 3 good
**Rating:** 6
**Confidence:** 4

**Summary:**

The focus of this work is to address the problem of task interference in multitask learning (MTL), which manifests as the negative effect that learning a task may have on another one when trained together. To this end, the authors propose a new soft parameter-sharing framework coined GatedMTL, which effectively consists of an automatic mechanism by which a series of identical task-specific architectures learn to share a mixture of their features during training, while retaining task-specific parameters when needed. The authors also propose to use sparsity regularization to encourage sharing parameters and reduce compute. Finally, empirical results on convolutional and transformer based models show that the proposed architecture is able to successfully explore the performance vs. compute trade-off, outperforming the chosen baselines in that matter.

**Strengths:**

- The paper is well-written, and the proposed solution is super intuitive and easy to understand.
- The emphasis on performance vs. flops (or size) is rather refreshing to read.
- The number of experiments variety is impressive for what is usual in the field, and it is nice to see a discussion and empirical evaluation of negative transfer and backbone size.
- The authors propose GatedMTL for two fairly widespread architectures, and the empirical results are quite positive.

**Weaknesses:**

**Limitations**
- W1. The biggest problem I have with the manuscript is that it does not discuss or show the limitations of the proposed approach _at all_, which can really easily mislead the readers (and thus, the reviewers). For example, to my understanding, the proposed approach at training time is $T$ individual models that are trained altogether. However, this is a _huge_ setback as it scales poorly with $T$ in memory and time (for example, the usual CelebA setting in MTL is to do a 40-task binary classification, but the authors reduce it to 3 tasks). The authors should discuss it in the manuscript and show training times for each of the experiments.
- W2. The hyperparameters $\tau_t$ are hardly intuitive, and the recommendation is to i) use the gap between STL and MTL models, and to ii) study the distribution of the gating patterns wrt the shared branch. The former requires tuning and training $T+1$ models, whereas the latter requires carefully looking into the model parameters. I am afraid that this can really hurt the adoption of the model by practitioners.

**Presentation**
- W3. Citations should properly use `\citet` and `\citep`. Even worse, the bibliography is a mess and I cannot comprehend how it happened (and I am going to assume, in good faith, that LLMs have nothing to do). The ones I spotted:
	- Kendall's citation is doubled (and with different years).
	- The citations **in the same paragraph of the manuscript** for DWA and MTAN (proposed in the same paper) are different. And again, different years. This is mind-blowing to me.
	- The paper by Maninis is also doubled.
	- The paper by Javaloy & Valera is from ICLR 2022, not 2021.
	- GradNorm is cited as arxiv 2017 when it is published at ICML 2018.
	- Adashare's paper has no venue.
	- Most urls point to semanticscholar instead of the official venue.

**Experiments**
- W3. I find $\Delta_{\text{MTL}}$ a brittle metric, as it is sensitive to low-magnitude metrics and task metrics are not comparable. I would add a more robust metric like the rank mean (see, e.g., [1]).
- W4. The chosen baselines are inconsistent across experiments and mostly outdated. From the MTO side, DWA and Uncertainty are quite old and weak in comparison with other methods like PCGrad, CAGrad, or NashMTL. From the side of adaptive architectures, more modern approaches like Adashare should be included.

[1] Navon, A., Shamsian, A., Achituve, I., Maron, H., Kawaguchi, K., Chechik, G., & Fetaya, E. (2022). Multi-task learning as a bargaining game. arXiv preprint arXiv:2202.01017.

**Questions:**

- Q1. Do you use a different Lagrange multiplier for each task when using L1 regularization? Otherwise, I don't see how it is comparable to the hinge loss in Eq. 4.

---

> ### Author Response · Authors · 2023-11-16
>
> We thank the reviewer for their thorough review and constructive feedback on our manuscript. We appreciate the recognition of the strengths of our work, finding our paper *well-written*, and the *proposed solution to be super intuitive*, as well as highlighting our *empirical evaluations to be impressive* and *the empirical results being quite positive*. While we understand the concerns raised regarding limitations, we would like to provide additional insights, clarifications, and further baseline experimentations to address the reviewer's feedback.
>
> **Training time:** The primary focus of GatedMTL is on inference-time efficiency, crucial in real-world applications. An example use case is XR applications requiring edge deployment for solving multiple tasks concurrently. We acknowledge that GatedMTL is not optimized for training time efficiency and we have added a limitation section in the paper, but we do not concur that this poses a significant drawback for practitioners. The training cost of GatedMTL is comparable, if not more efficient, than most existing MTO approaches. During training, many MTO approaches require $T$ backward passes and additional computations such as computing gradient conflicts/storing per task gradients simultaneously. GatedMTL, however, allows for sparsification into both forward and backward passes, meaning parameters that are not chosen do not get gradients and hence enables significant memory and compute saving (depending on the sparsity level). At inference, where GatedMTL is primarily designed for, we outperform all competing MTO baselines offering a significantly better performance-compute trade-off.
>
> **Scaling to an extremely high number of tasks:** We agree that GatedMTL may not be the optimal choice for scenarios requiring simultaneous training/inference for an exceptionally high number of tasks, such as 40. However, we observe that this limitation is not unique to GatedMTL but is a common challenge across the MTL field. Our framework, therefore, remains relevant and effective within the practical range of multi-task learning scenarios that the vast majority of the related literature is focusing on.
>
> **Hyperparameter $\tau$:** We believe setting of this parameter to be both intuitive and straightforward. A lower value indicates more reliance on shared features, whereas a higher value suggests a need for more task-specific ones. We would like to ask for clarification of what the reviewer finds unintuitive about $\tau$? In addition, in MTL research and practice, establishing single-task baselines is a standard practice to gauge the relative performance of MTL models. Therefore, having access to single task models is not unreasonable from a practical perspective. In addition, as shown in Appendix C the parameter $\tau$ itself is very robust.
>
> **Citations:** We acknowledge the issues with our citations and bibliography and have meticulously revised this section, ensuring that all references are accurately and consistently cited.

---

> ### Author Response · Authors · 2023-11-16
>
> **Mean Rank Metric:** We thank the reviewer for this suggestion. We added the rank mean measure to all tables in the paper and updated the manuscript. This new metric further strengthens the merits of GatedMTL, consistently showing strong performance across both the $\Delta_{MTL}$ and rank mean measures.
>
>
> **Baselines:** In response to the reviewers feedback regarding the selection of baselines, we have extensively expanded our experiments to include the suggested state-of-the-art methods, i.e. PCGrad [1], CAGrad [2], RDW [3], and MGDA [4]. The results are presented in the table below. As can be seen, GatedMTL substantially outperforms all MTO baselines across all the metrics. We also integrated the implementation of NashMTL into our MTL baselines, however, despite extensive hyperparameter search, the experiments diverged which are not reasonable to be reported.
>
> Updated performance comparison on NYUD-v2 using HRNet-18 backbone. The new baselines are shown in bold.
> Model|Semseg|Depth|Normals|$\Delta_\text{MTL}$(\%)|Flop(G)|MR
> -|-|-|-|-|-|-
> STL|41.70|0.582|**18.89**|0|65.1|8.0
> MTL(Uni.)|41.83|0.582|22.84|-6.86|24.5|11.0
> DWA|41.86|0.580|22.61|-6.29|24.5|8.7
> Uncertainty|41.49|0.575|22.27|-5.73|24.5|8.3
> Auto-$\lambda$|42.71|0.577|22.87|-5.92|24.5|7.7
> **RDW**|42.10|0.593|23.29|-8.09|24.5|11.7
> **PCGrad**|41.75|0.581|22.73|-6.70|24.5|10.3
> **MGDA**|41.23|0.625|21.07|-6.68|24.5|11.3
> **CAGrad**|42.31|0.580|22.79|-6.28|24.5|8.7
> GatedMTL|**43.58**|**0.557**|19.32|**+2.18**|43.2|**1.3**
> GatedMTL|42.93|0.5613|19.72|+0.70|38.4|2.3
> GatedMTL|42.43|0.5671|20.01|-0.54|36.0|4.0
> GatedMTL|42.85|0.5778|21.01|-2.58|33.0|5.0
> GatedMTL|42.35|0.5755|21.70|-4.06|29.1|6.0
>
> ----------
> Updated performance comparison on NYUD-v2 using ResNet-50 backbone.
>
> Model|Semseg|Depth|Normals|$\Delta_\text{MTL}$(\%)|Flops(G)|MR
> -|-|-|-|-|-|-
> STL|43.20|0.599|**19.42**|0|1149|9.0
> MTL(Uni.)|43.39|0.586|21.70|-3.04|683|9.7
> DWA|43.60|0.593|21.64|-3.16|683|9.7
> Uncertainty|43.47|0.594|21.42|-2.95|683|10.0
> Auto-$\lambda$|43.57|0.588|21.75|-3.10|683|10.0
> **MGDA**|42.56|0.586|21.76|-3.83|683|11.3
> **RDW**|43.49|0.587|21.54|-2.74|683|8.3
> **PCGrad**|43.74|0.588|21.55|-2.66|683|7.3
> **CAGrad**|43.57|0.583|21.55|-2.49|683|7.0
> MTAN|**44.92**|0.585|21.14|-0.84|683|4.0
> Cross-stitch|44.19|0.577|19.62|+1.66|1151|2.3
> GatedMTL|44.15|**0.573**|19.49|**+2.08**|916|**2.0**
> GatedMTL|43.70|0.578|19.65|+1.16|892|4.0
> GatedMTL|42.99|0.589|19.93|-0.48|798|9.7
>
> Please note that the MTL setting we are targeting, solves multiple tasks *simultaneously in one forward pass*. In contrast, Adashare and existing MoE-based approaches, can only solve one task at a time, requiring one forward pass per task.
> Therefore, these models are not suitable for the setup we are targeting in this work.
>
> **Q) Different Lagrange multipliers for L1 regularization:**
> Yes, we indeed experimented with varying Lagrange multipliers for the L1 sparsity term for each task. However, we observed that the Lagrangian multipliers did not yield similar training dynamics for the gates as compared to the hinge loss. We hypothesize that, without a hinge target rate, the sparsification endlessly continues throughout the training making it difficult for the model to converge and the task-specific and shared representations to co-adapt.
>
> We trust that we have addressed all the comments raised by the reviewer and are happy to further clarify any points as needed.
>
> References:
>
> [1] Gradient Surgery for Multi-Task Learning. NeurIPS, 2020.
>
> [2] Conflict-Averse Gradient Descent for Multi-task Learning. NeurIPS, 2021.
>
> [3] Reasonable Effectiveness of Random Weighting: A Litmus Test for Multi-Task Learning. TMLR, 2022.
>
> [4] Multi-Task Learning as Multi-Objective Optimization. NeurIPS, 2018.

---

> > ### Comment · Reviewer_wdHJ · 2023-11-20
> >
> > Dear authors, thank you for the review and the extra baselines. Let me answer and clarify some of the points of the review:
> >
> > **Training and scaling** Say that I buy the argument provided in the rebuttal, which seems sensible at first. Irrespectively of if I buy it, discussing efficiency and _showing_ training times in the paper is crucial. Besides, if training performance is comparable to some MTO methods, reducing the 40-task CelebA used in those papers to a 3-task CelebA does nothing but rising an eyebrow of the reader.
> >
> > **Hyperparameter** Higher $\tau$ implies more task-specific parameters is not what I would call intuitive. What I find unintuitive is that I cannot find an easy interpretation of the parameter so that a practitioner can tune it. If $\tau$, say, meant the desired percentage of task-specific parameters for a task, then it would have a clear interpretation that I can reason about, but instead it is recommended to look at T models and the distribution of activations to tune them.
> >
> > I also do not concur with that having single-task models is a standard practice. And, if that were the case, why would one throw away all the information about the STL weights for the MTL model?
> >
> > **Baselines** Thanks for the new results. I have one question with respect to MGDA: do you apply MGDA on the parameters of the backbone or on the output of the backbone? If it is the former, that is MGDA as by Desideri, if it is the latter, that is the MGDA-UB algorithm proposed by Sener ([4] in the references above) to precisely avoid running more than 1 backpropagation call.
> >
> > If the manuscript has been updated with new sections, I'd like to note to the authors that ICLR allows uploading revisions of the manuscript during the rebuttal (see the [guide for authors](https://iclr.cc/Conferences/2024/AuthorGuide)).

---

> ### Author Response · Authors · 2023-11-22
>
> We thank the reviewer for their response and clarifying these points further.
>
> **Training and scaling:** In response to the reviewer's concern about training time efficiency, we report a comprehensive benchmarking of our GatedMTL model against various MTL approaches on our most complex dataset, Pascal-Context with 5 tasks. We include the average time for a single iteration for forward pass, backward pass, as well as the training time. As observed in the table below, while GatedMTL is generally cheaper than MTO approaches and Cross-Stitch, it adds non-substantial training overhead compared to standard MTL yielding a much higher $\Delta_\text{MTL}$ measure. For instance, our GatedMTL variant with a training time of 8.4h adds 11\% to the training time of standard MTL while improving the  $\Delta_\text{MTL}$ from -4.14 to -1.35. More details about the setup of the experiments are provided on Appendix F of the paper.
>
> **Method**|**Forward (ms)**|**Backward (ms)**|**Training time (h)**|**$\Delta_\text{MTL}$**
> -|-|-|-|-
> **Standard MTL**|60|299|7.5|-4.14
> **MTAN**|73|330|8.5|-1.78
> **Cross-stitch**|132|454|12.3|+0.14
> **MGDA-UB**|60|568|13.2|-1.94
> **CAGrad**|60|473|11.1|-2.03
> **PCGrad**|60|495|11.6|-2.58
> **GatedMTL**|76|324|8.4|-1.35
> **GatedMTL**|102|376|10.1|+0.12
> **GatedMTL**|119|426|11.5|+0.42
>
> With regards to experiments on CelebA, we considered this as a toy classification dataset, particularly for some of our ablation studies on the capacity experiments. However, we are fine with moving the CelebA materials to the appendix if the reviewer recommends that.
>
> **Hyperparameter $\tau$:** The reviewer suggests that if $\tau$ represented "*the desired percentage of task-specific parameters for a task*", it would offer a more intuitive and practical understanding for tuning. We would like to clarify that this is indeed exactly the intended interpretation of the $\tau$ parameter, as implemented in our method.
>
> As detailed in Equation (4), the 'target rate' is designed to control the proportion of active gates at each specific layer, with each gate governing the activation of a single convolutional kernel. Thus, $\tau$ sets a soft upper limit for active task-specific parameters through these gates.
>
> Accordingly, we have edited the manuscript to further clarify this.
>
> **Using STL weights:** We respectfully disagree with the assertion that utilizing single-task model (STL) weights should not be practiced in multi-task learning (MTL). Our position is that the inability of existing MTL solutions to effectively incorporate STL weights mainly because of inherent design limitations, represents a notable limitation in the field. This should not, however, be a deterrent to the development and adoption of innovative approaches that can successfully leverage STL weights, as we propose in our work.
>
> We consider our method’s ability to harness STL weights for enhancing the performance of MTL models during inference as a strength and a novel aspect of our research. Far from being a questionable practice, employing STL data in this manner aligns with practical machine learning principles and optimizes the use of available resources.
>
> In addition, in numerous real-world applications, the gains in efficient inference over time, especially when scaled across millions of edge devices, can significantly offset the additional requirement of collecting STL weights. We would like to emphasize that single task pre-trained models for tasks such as semantic segmentation, depth estimation, etc. are abundantly available for a multitude of model architectures that could be plugged-in off-the-shelf.
>
> However, to ensure full transparency with readers, we have acknowledged the necessity of collecting STL weights in the limitations section of our manuscript to clearly inform readers of any potential trade-offs involved in our approach.
>
> **Baselines:** We are indeed using MGDA-UB algorithm proposed by Sener and now call the algorithm accordingly in our tables.
>
> We are are attaching an updated version of our manuscript by highlighting the changes in blue.

---

> > ### Comment · Reviewer_wdHJ · 2023-11-23
> >
> > While I still don't fully agree with some of the argumentation during the rebuttal, I think the authors have made significant effort to address the points raised during the discussion to improve the state of the manuscript.
> >
> > Therefore, I have raised my score from 3 to 6.

---

### Author Response · Authors · 2023-11-22

We would like to thank all the reviewers for their thorough reviews and insightful feedback which we used to significantly improve our paper. We are highly encouraged that the reviewers found our proposed method *"original"* (**PvDp**) and *"novel"* (**7nfe**), *"well written"* (**wdHJ**, **PvDp**), *"super intuitive"* (**wdHJ**), and *"easy to understand"* (**wdHJ**, **DXQP**). We are pleased that three of the reviewers appreciated our experiments to be *"extensive"* (**wdHJ**, **PvDp**, **7nfe**), with *"impressive variety"* (**wdHJ**), while describing our empirical results *"quite positive"* (**wdHJ**), with *"obvious improvements"* (**PvDp**), and demonstrating *"competitive performance"* (**DXQP**).

We have thoroughly addressed each reviewer's individual concerns separately through comments, and we would like to use this global response to highlight the additional extensions to our empirical evaluations:

**Addition of new baselines and metrics:** We extensively expanded our experiments to include four new state-of-the-art baselines, i.e. PCGrad, CAGrad, MGDA-UB, and RDW. The updated results are provided in Tables 1 and 2 for NYUD-v2 and Table 5 for Pascal-Context. We additionally report the mean rank (MR) evaluation metric alongside $\Delta_\text{MTL}$ in the paper. GatedMTL consistently and significantly outperforms all competing methods in both existing and the new MR metrics. Finally, we also report the parameter count for our method and baselines.

**Total training, forward pass, and backward pass timing:** We report the average time for a single iteration for forward pass, backward pass, as well as the total training time in Table 9 in the Appendix. While GatedMTL is generally cheaper to train than MTO approaches and Cross-Stitch, it adds non-substantial training overhead compared to standard MTL yielding a much higher $\Delta_\text{MTL}$ measure. For instance, our GatedMTL variant with a training time of 8.4h adds 11\% to the training time of standard MTL while improving the  $\Delta_\text{MTL}$ from $-4.14$ to $-1.35$.

**Clarifications and limitations section:** We added several improvements to further clarify the method and hyperparameters, e.g. a detailed pseudo code for the forward pass of the GatedMTL encoder, distinction with MoE-based methods, and choice of scalarization weights. Furthermore, we added a limitations section at the end of the paper addressing points such as training time, STL weights, scaling to extremely high number of tasks, and targeting a specific computational cost.

**Updated manuscript:** We have submitted an updated version of the manuscript, incorporating the aforementioned improvements. For ease of review, we have highlighted these changes in blue.

---

### Meta-Review · Area_Chair_kTBK · 2023-12-11

**Metareview:**

In this paper, the authors proposed to use the differentiable gate to select and combine channels from its task-specific features and a shared memory bank of features for multi-task learning setting. The authors conducted empirical comparison to justify the benefits of the proposed method.

**Justification For Why Not Higher Score:**

However, there are several concerns raised by the reviewers.

1, In fact, the differentiable gate has been proposed and used in many problems for different purpose, which diminishes the novelty of the proposed method.

2, The place in encoder or decoder to add differentiable gates seems arbitrary. It will be great the authors can add more discussion and ablation about this issue for the completeness.

3, The writting of the paper should be imprved to resolve the confusion from the reviewers, instead replying through rebuttal.

**Justification For Why Not Lower Score:**

N/A

---

### Decision · Program_Chairs · 2024-01-16

Reject